# TssA–TssM–TagA interaction modulates type VI secretion system sheath-tube assembly in *Vibrio cholerae*

Maria Silvina Stietz[1], Xiaoye Liang[1,2], Hao Li[2], Xinran Zhang[1] & Tao G. Dong [1,2✉]

The type VI protein secretion system (T6SS) is a powerful needle-like machinery found in Gram-negative bacteria that can penetrate the cytosol of receiving cells in milliseconds by physical force. Anchored by its membrane-spanning complex (MC) and a baseplate (BP), the T6SS sheath-tube is assembled in a stepwise process primed by TssA and terminated by TagA. However, the molecular details of its assembly remain elusive. Here, we systematically examined the initiation and termination of contractile and non-contractile T6SS sheaths in MC-BP, *tssA* and *tagA* mutants by fluorescence microscopy. We observe long pole-to-pole sheath-tube structures in the non-contractile MC-BP defective mutants but not in the Hcp tube or VgrG spike mutants. Combining overexpression and genetic mutation data, we demonstrate complex effects of TssM, TssA and TagA interactions on T6SS sheath-tube dynamics. We also report promiscuous interactions of TagA with multiple T6SS components, similar to TssA. Our results demonstrate that priming of the T6SS sheath-tube assembly is not dependent on TssA, nor is the assembly termination dependent on the distal end TssA–TagA interaction, and highlight the tripartite control of TssA–TssM–TagA on sheath-tube initiation and termination.

[1] Ecosystem and Public Health; Snyder Institute for Chronic Diseases; Biochemistry and Molecular Biology, Cumming School of Medicine, University of Calgary, Calgary, AB T2N4Z6, Canada. [2] State Key Laboratory of Microbial Metabolism, Joint International Research Laboratory of Metabolic & Developmental Sciences, School of Life Sciences and Biotechnology, Shanghai Jiao Tong University, Shanghai 200240, China. ✉email: tdong@ucalgary.ca

The type VI secretion system (T6SS) is a widespread weapon employed by many Gram-negative bacteria to survive in the gut microbiome and other competitive environments[1–4]. Evolutionarily related to bacteriophage tails, the T6SS is composed of a trans-membrane complex (MC), a cytoplasmic baseplate (BP), and a long double tubular sheath-tube structure (Fig. 1a)[5–9]. Sheath contraction delivers the inner Hcp tube and its tip spike VgrG–PAAR complex and associated effectors to the extracellular environment or directly into a recipient cell[10,11]. Depending on effector functions, the T6SS can intoxicate both eukaryotic and bacterial cells[1,12–14]. The contracted sheath is disassembled by an AAA+ family ATPase ClpV for recycling[15–18].

Biogenesis of the T6SS has been proposed to follow a sequential order that initiates from MC assembly, followed by BP recruitment and the extension of sheath-tube[6,10]. Firstly, the outer-membrane lipoprotein TssJ recruits the inner-membrane protein TssM that binds to another inner membrane partner TssL, forming a 5-fold trans-envelope MC composed of 15 copies of TssJ and 10 copies each of TssM and TssL[8]. The BP comprises 6 copies of the TssEFGK wedge complex wrapping around a trimeric VgrG spike in the center, with TssK serving as the main connector to the TssML inner membrane complex and TssF

interacting with VgrG and the sheath VipAB (also known as TssBC)[9,19,20]. The tip protein PAAR is essential for T6SS functions in *Acinetobacter baylyi* ADP1 but not strictly required in *Vibrio cholerae*[21]. Interestingly, while the BP and sheath-tube components are evolutionarily related to contractile phage-tail proteins[3,22,23], the inner-membrane MC proteins TssM and TssL are homologues of the type IV secretion system proteins IcmF and DotU, respectively[6,24,25]. Finally, the sheath–tube extends from the BP by assembling protomers at the distal end and can span across the cytoplasm prior to contraction[11]. What triggers the contraction of the extended sheath remains elusive, but insertions in the VipA N-terminal linker region can block contraction and result in non-contractile sheath-tube structures locked in the pre-contraction state[20,26].

Of all T6SS-associated proteins, proteins with the conserved ImpA_N domain remain the least understood due to their multiple binding partners and sequence divergence among species[27–31]. These proteins feature a conserved N-terminal domain (ImpA_N) and a divergent C-terminal domain (CTD), and have been categorized into several clades that exhibit different symmetry of the CTD oligomers, including a 6-fold dodecamer, a 16-fold 32-subunit ring, and a 5-fold decamer[27,28]. Recently the

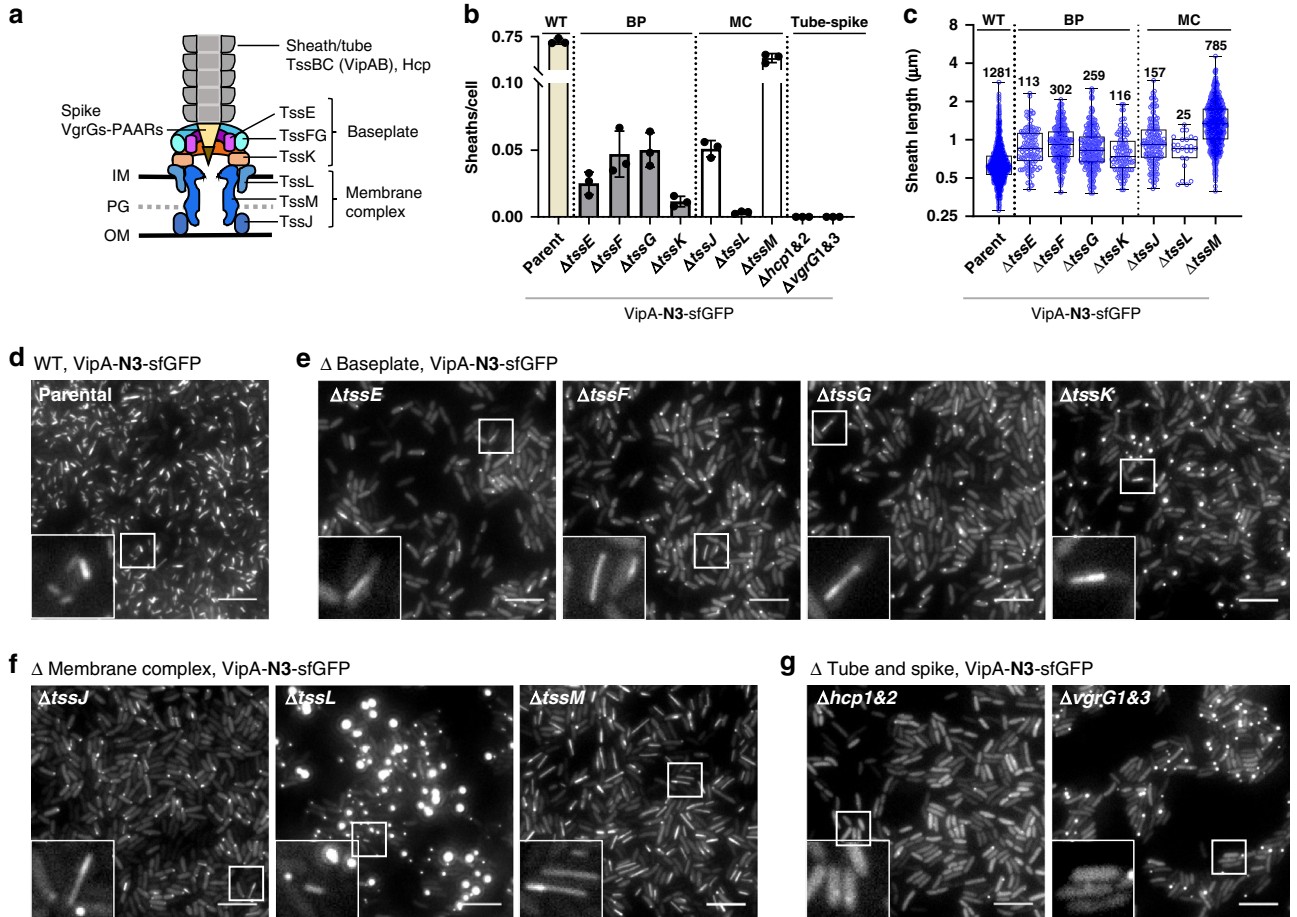

**Fig. 1 T6SS sheaths assemble in membrane complex and baseplate mutant strains. a** Model of T6SS illustrating the structural components. **b** Number of non-contractile sheaths over total cells is indicated for parental (WT) and for each deletion strain (BP, baseplate; MC, membrane complex and Tube-Spike). Each data point represents an independent biological replicate (n = 3). Error bars represent the mean value ± SD of three biological replicates. **c** Box plot shows the non-contractile sheath length (in µm) in the parental and deletion strains of the BP and MC components. Whiskers show Min to Max values and boxes indicate median and extend from the 25th to 75th percentiles. Each data point represents a sheath, n is indicated in the graph. Data were acquired from three independent experiments. Y axis shows Log2 scale. Cropped images (30 ×30 µm) correspond to sfGFP labeled non-contractile VipA sheaths (VipA-N3-sfGFP) assembled in: parental strain (**d**) or deletion strains of baseplate proteins TssE, TssF, TssG, and TssK (**e**); membrane complex components TssJ, TssL, and TssM (**f**) and double deletion of Hcp1&2 or VgrG1&3 (**g**). All images are representative of at least three independent experiments. Scale bars, 5 µm. Insets (3.2 ×3.2 µm) in bottom left correspond to areas indicated by white boxes. GFP channel is shown in grayscale.

ImpA_N proteins were classified into three functional classes: distal end chaperone that facilitates polymerization (TssA/TsaC), membrane anchor that terminates polymerization (TagA/TsmA), and baseplate component for assembly (TsaB)[29]. The T6SS cluster of *V. cholerae* encodes two ImpA_N domain proteins, TssA and TagA, which belong to the TsaC and TsmA classes, respectively. It has been shown in *Escherichia coli* and *V. cholerae* that TssA (TsaC) could interact with the inner membrane proteins TssL and TssM, the baseplate TssEFGK-VgrG, TagA (TsmA) and subunits of sheath-tube VipB (TssC) and Hcp[28–30], playing a key role in priming and facilitating sheath-tube polymerization[28–30]. TagA (TsmA) is peripherally associated with the inner membrane and is believed to function as a stopper that terminates sheath extension upon interacting with TssA at the sheath distal end[30,32]. Deletion of *tagA* results in frequent formation of long and curved sheath-tube in both *E. coli* and *V. cholerae*[30,32,33]. The H1-T6SS cluster of *Pseudomonas aeruginosa* encodes a TsaB (TssA1) that is a baseplate component and interacts with TssK1, TssF1, ClpV1, and sheath-tube subunits VipA1(TssB1) and Hcp1[31]. Despite of the structural and functional differences of these ImpA_N domain proteins, it seems to be common that they have multiple binding partners and their functions remain to be elucidated. Although it is known that individual components of the MC-BP complex play critical roles in T6SS assembly, it remains unclear how each component contributes to the stepwise initiation and extension processes. In this study, we aim to address this question by comparing sheath assembly of wild type contractile sheath and mutant non-contractile sheath using a series of MC-BP, TssA, and TagA mutants in *V. cholerae*. We have made a number of observations that cannot be fully accounted for by the current model of T6SS assembly, especially regarding the priming of sheath assembly. First, we found that non-contractile sheath structures were assembled in the Δ*tssM* and other MC-BP deletion mutants indicating that the initiation of sheath-tube assembly does not require a fully assembled MC-BP. Moreover, deletion of *tagA* and overexpression of TssM could significantly stimulate wild-type-like sheath assembly in the otherwise T6SS-inactive Δ*tssA* mutant, which argues against the previously proposed role of TssA as an essential component for priming the assembly and for terminating the assembly at the distal end by interacting with TagA[28]. Lastly, overexpression of TssA stimulated non-contractile sheath formation in TssA+ cells but inhibited sheath formation in the Δ*tssM* mutant, and the inhibition can be relieved by co-expressing VgrG. These data not only reveal that an intact MC-BP is dispensable for priming sheath polymerization but also places TssA–TssM–TagA in the center of regulating sheath-tube initiation and termination through their complex interactions with multiple structural components.

## Results

**Non-contractile T6SS structures assemble in MC-BP mutants**. MC-BP mutants have been previously shown to be defective in T6SS-mediated bacterial killing in *V. cholerae*[34]. The loss of T6SS functions could result from the inability to initiate structural assembly or to maintain the stability of polymerizing structures. To distinguish the contribution of each MC-BP component to sheath assembly and stability, we employed a previously established non-contractile sheath model with a 3 amino acid (Ala-Glu-Val) insertion in the VipA(TssB) N-terminal linker region[26] and chromosomally introduced this construct, hereafter referred to as VipA-N3-sfGFP, to a panel of individual MC-BP mutants. Using fluorescence microscopy imaging of sheath assembly, we found that non-contractile but otherwise wild type cells formed frequent sheath-tube structures of average length (0.68 μm)

(Fig. 1b–d) comparable to their contractile counterparts (Supplementary Fig. 1d), which is consistent with previous findings[26]. Surprisingly, long and often pole-to-pole sheath structures were found in all non-contractile MC-BP mutants (Fig. 1c, e, f), most abundantly present in the inner-membrane MC gene mutant Δ*tssM* and about 10-fold fewer in the other MC-BP mutants (Fig. 1b, c). No sheath was observed in the Δ*hcp1&2* and the Δ*vgrG1&3* mutants indicating non-contractile sheath-tube polymerization is not a spontaneous process but dependent on Hcp and VgrG1&3 (Fig. 1b, g). Similar to Δ*vgrG1&3* mutant, BP mutants and the outer-membrane MC gene Δ*tssJ* mutant exhibited predominantly polar foci that resemble inclusion bodies of protein aggregates[35] (Fig. 1e–g). Notably, the other inner-membrane MC gene mutant Δ*tssL* showed larger foci that are absent in the Δ*tssM* strain, indicating that TssM and TssL play distinct functions besides being the building blocks of the inner membrane complex. Considering that the Δ*tssM* mutant is similar to TssM+ in the number of sheath-tube formed (Fig. 1b), the assembly of much longer structures in the Δ*tssM* suggests that TssM is important for termination but not initiation of T6SS sheath-tube assembly. For simplicity, we named the long and pole-to-pole sheath-tube structures LPPS hereafter. Notably, initiation of these LPPS structures is not limited to the pole area but appears to occur randomly throughout the cell (Supplementary Fig. 1g, h and Supplementary Movie 1).

For comparison, we also examined contractile sheath formation in these MC-BP mutants. All mutants exhibited variable levels of polar foci, indicative of sheath-aggregation, with the Δ*tssE* and the Δ*tssG* showing the lowest level (Supplementary Fig. 1a–c, f). Similar to the non-contractile model, the Δ*tssL* mutant showed larger foci that are absent in the Δ*tssM* mutant (Supplementary Fig. 1b). However, no MC-BP mutant exhibited contractile sheath-tube assembly (Supplementary Fig. 1a–c). Cumulatively, these findings demonstrate that T6SS non-contractile sheaths can assemble in the membrane complex and baseplate mutant strains.

**TssM controls T6SS sheath-tube termination**. The abundant LPPS observed in non-contractile Δ*tssM* cells led us to investigate how TssM affects sheath-tube termination. First, we tested in contractile sheath cells whether the *tssM* deletion can be complemented to eliminate any polar effect on its neighboring *tssA* and *tagA* genes, shown to be important for priming and termination, respectively[28,30]. Deletion of *tssM* expectedly abolished assembly of the contractile sheath and the *E. coli* killing ability, which can be complemented by ectopic expression of TssM (Fig. 2a, b). Notably, upon pTssM induction, LPPS structures were observed in contractile cells (Fig. 2a, Supplementary Movie 2).

Similarly, in non-contractile sheath cells, LPPS structures were not only formed in the Δ*tssM* strain but also, albeit less frequently, in parental TssM+ cells overexpressing pTssM (Fig. 2c, d). Once LPPS is formed in the Δ*tssM* mutant, ectopic expression of TssM could not reduce LPPS to parental TssM+ levels (Fig. 2c). These data provide further evidence supporting that TssM plays a previously unrecognized role in controlling termination, besides serving as a MC building block. Specifically, LPPS formation induced by overexpression of TssM may result from TssM sequestering other proteins involved in T6SS termination.

**TssA inhibits sheath assembly in the MC-BP mutants**. Since TssA is believed to be key in priming sheath assembly, we next determined how TssA affects LPPS formation. Interestingly, induction of TssA synthesis in the non-contractile VipA-N3-

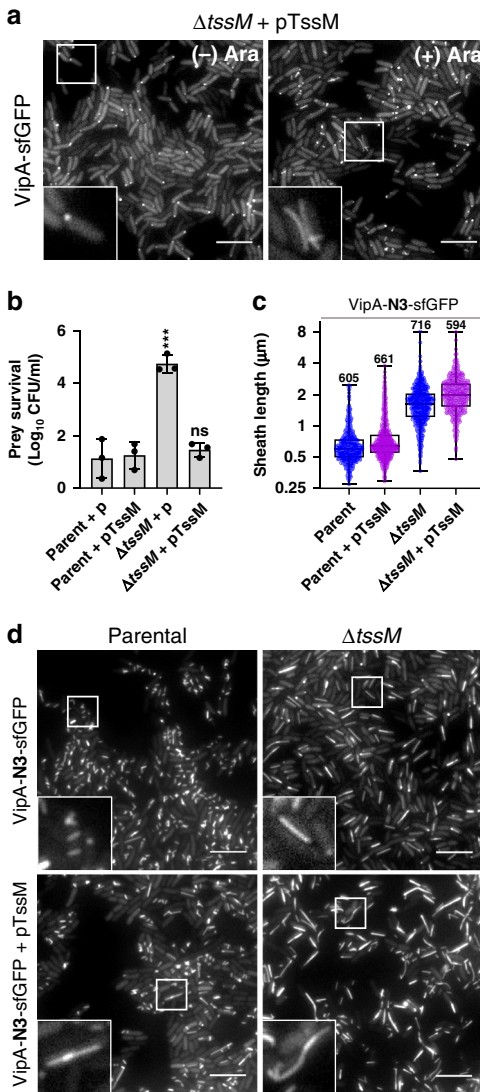

**Fig. 2 Overexpression of TssM leads to LPPS formation. a** Still Images (30 ×30 μm) of contractile VipA-sfGFP Δ*tssM* strain complemented with pTssM plasmid (Δ*tssM*/pTssM) without ((−) Ara, left) or with 0.1% L-arabinose induction ((+)Ara, right). **b** Relative survival of *E. coli* prey after co-incubation with parental (WT) or Δ*tssM* killer cells carrying pBAD plasmid, empty (p) or expressing *tssM* (pTssM). One-way ANOVA with Sidak's multiple comparisons test comparing each strain to the parental strain carrying the same plasmid, ***$p < 0.001$; ns, not significant. Data points represent independent experiments ($n = 3$). Error bars indicate the mean value ± SD of three biological replicates. **c** Box plot shows sheath length, in μm, measured in non-contractile parental and Δ*tssM* strain and in the same strains overexpressing pTssM plasmid after induction with 0.4% L-arabinose. Whiskers show Min to Max values and boxes indicate median and extend from the 25th to 75th percentiles. Each data point represents a sheath, $n$ is indicated in graph for each strain. *Y* axis shows Log2 scale. Two-tailed Mann–Whitney test comparing each strain with and without overexpression of TssM was performed ($p < 0.001$). **d** Images of non-contractile VipA-N3-sfGFP parental and Δ*tssM* strain (Top) and strains with pTssM plasmid overexpression (Bottom). In (**a**, **d**) images are representative of at least three independent experiments, insets (3.2 × 3.2 μm) in bottom left correspond to area in white boxes. Scale bars, 5 μm. GFP channel is shown in grayscale.

sfGFP cells doubled the sheath length and stimulated the formation of LPPS (Fig. 3a, b), resembling the ones observed in Δ*tssM* cells. This might result from the known promiscuous binding capabilities of TssA to T6SS components including TssM or TagA, which in turn impair sheath-tube termination. Surprisingly, when induced in the non-contractile Δ*tssM* mutant, expression of TssA abolished sheath-tube formation (Fig. 3c). Because the VgrG–PAAR spike complex is required for assembly, we hypothesize such inhibition is due to sequestration of VgrG or PAAR by TssA. Therefore, we tested co-expression of TssA with VgrG1 or VgrG3 as well as PAAR2 in the Δ*tssM* mutant. Assembly of LPPS was partially restored under co-expression with either of the VgrGs but not with PAAR2 or an empty vector (Fig. 3d, e). We also tested TssA overexpression in the BP mutants. Interestingly, induction of TssA effectively reduced sheath and foci formation in Δ*tssE*, Δ*tssF*, Δ*tssG*, and Δ*tssK*, suggesting TssA prevents aggregation of sheath protomers in BP-defective mutants (Supplementary Fig. 2a, b). Cumulatively, these findings suggest that overexpressed TssA sequesters VgrG to inhibit sheath assembly, but intact MC-BP can disassociate TssA-bound VgrG to circumvent this inhibition.

**ImpA_N domain is dispensable for sheath-tube initiation.** We next examined the effect of the conserved ImpA_N domain of TssA (A11-Q126) and TagA (T16-N118) on sheath-tube assembly (Supplementary Fig. 3a). Deletion of *tssA* or its ImpA_N domain (Δ*tssA*$^N$) in contractile cells abolished T6SS killing activities against *E. coli* (Fig. 4a). Assembly of T6SS was undetectable in the contractile Δ*tssA* mutant but a small number (5% of TssA$^+$ level) of sheath-tube was detected in the Δ*tssA*$^N$ mutant (Fig. 4b, d, h, Supplementary Movie 3). Complementation with plasmid-borne *tssA* restored killing ability and sheath assembly in both mutants (Fig. 4a–d, h). We then imaged the non-contractile VipA-N3-sfGFP sheath mutants. While the Δ*tssA* mutant showed only a few structures, the Δ*tssA*$^N$ mutant showed abundant wild-type-like sheath structures (Fig. 4c, e), suggesting the ImpA_N domain is dispensable for sheath-tube initiation. When TssA is overexpressed in non-contractile Δ*tssA* and Δ*tssA*$^N$ cells, we observed the formation of LPPS (Fig. 4c, e and Supplementary Fig. 3b) similar to its effect on non-contractile TssA$^+$ cells (Fig. 3a, b).

We next imaged contractile sheath formation in the Δ*tagA* and the Δ*tagA*$^N$ mutants, the latter lacking the ImpA_N domain. In these VipA-sfGFP contractile cells, both wild-type-like and LPPS structures were formed, and competition assays show that Δ*tagA*$^N$ mutant retained killing ability similar to Δ*tagA* cells (Fig. 4f, g and Supplementary Movie 4). Also, as previously reported[29,33], complementation with an inducible plasmid-borne *tagA* reduced the killing effectiveness as well as T6SS assembly in both TagA$^+$ and the Δ*tagA* mutants (Fig. 4f and Supplementary Fig. 3c). Cumulatively, the ImpA_N domain is nonessential for T6SS sheath-tube initiation.

**TagA and TssM control sheath-tube assembly in Δ*tssA*.** Because of the inhibition of assembly of T6SS sheaths by TagA overexpression (Supplementary Fig. 3c) and the interaction between TagA and TssA[29,30], we hypothesize that the impaired T6SS assembly in the Δ*tssA* mutant might result from the inhibitory effect of TagA. Indeed, when we introduced *tagA* deletion into the Δ*tssA* mutants in both contractile and non-contractile models, T6SS sheath formation was significantly increased (Fig. 5a, d–g and Supplementary Movie 5). However, there were still 20-fold

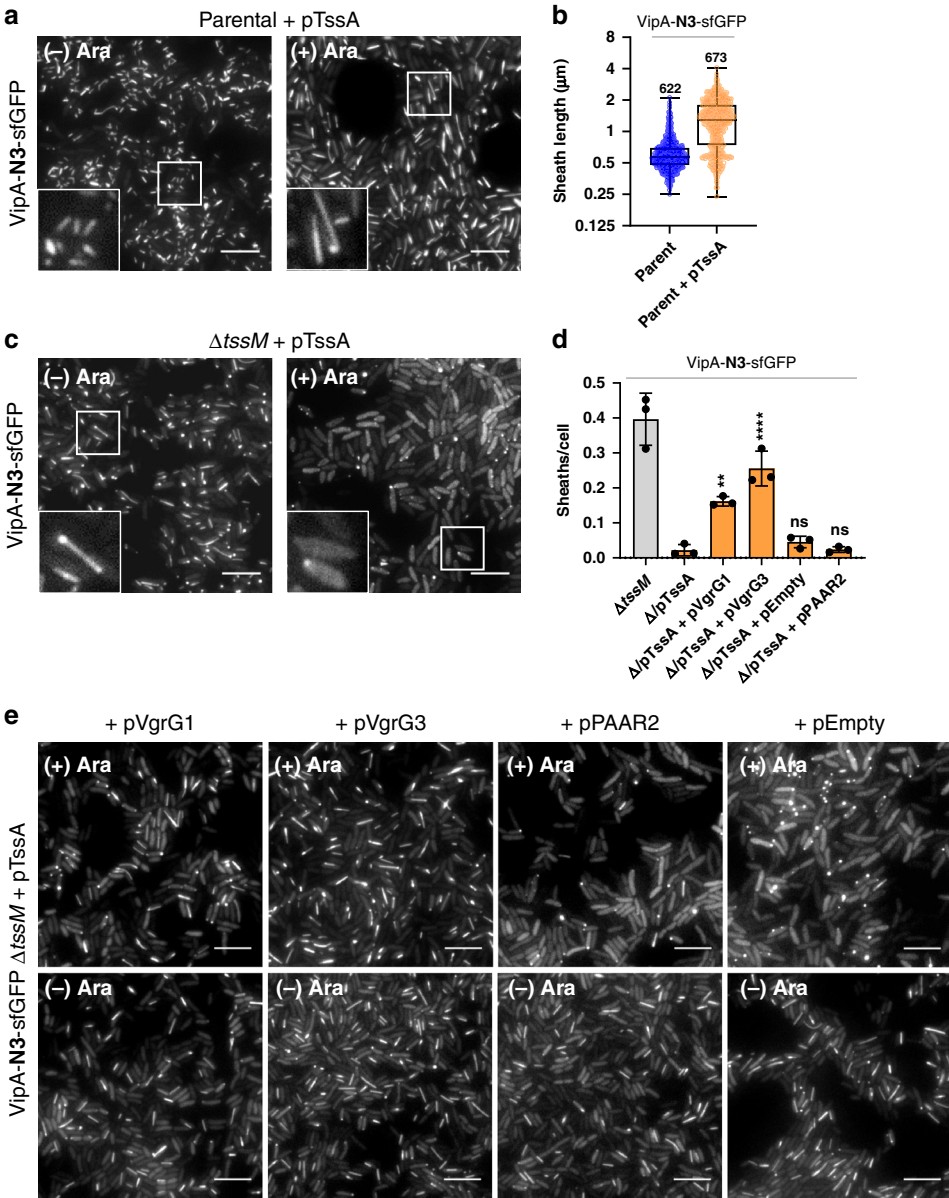

**Fig. 3 Overexpression of TssA sequesters VgrG to inhibit sheath assembly in the absence of TssM. a** Sheaths assembled in the non-contractile VipA-N3-sfGFP parental strain carrying plasmid pTssA without (−) or with (+) 0.4% L-arabinose induction. **b** Sheath length, in μm, comparing non-contractile parental strain with or without overexpression of pTssA. Whiskers show Min to Max values and boxes indicate median and extend from the 25th to 75th percentiles. Each data point represents a sheath (*n* is indicated in graph). Data were acquired in three independent experiments. *Y* axis shows Log2 scale. **c** Images show Δ*tssM* in the non-contractile VipA-N3-sfGFP strain before (left) and after (right) overexpression of pTssA vector. **d** Bar graph shows number of sheaths over total cells assembled in the VipA-N3-sfGFP Δ*tssM* strain without plasmid (Δ*tssM*), carrying only pTssA plasmid (Δ/pTssA), or co-expressing: TssA and VgrG1 (Δ/pTssA+pVgrG1), TssA and VgrG3 (Δ/pTssA+pVgrG3), TssA and empty vector (Δ/pTssA+pEmpty), or TssA and PAAR2 (Δ/pTssA+pPAAR2). One-way ANOVA with Sidak's multiple comparisons test comparing strain carrying pTssA only to strains carrying two plasmids, **$p < 0.01$, ****$p < 0.0001$, ns, not significant. Each data point represents an independent experiment (*n* = 3). Error bars indicate the mean value ± SD of three biological replicates. **e** Images of the VipA-N3-sfGFP Δ*tssM* strain expressing pTssA in combination with pVgrG1 or pVgrG3 or the control strains expressing pTssA with pPAAR2 or empty vector. All strains were induced with 0.4% L-arabinose for 2.5 h. Scale bars, 5 μm.

(contractile) and 6-fold (non-contractile) less than the corresponding parental strains and killing ability was not restored in the T6SS contractile cells (Fig. 5a, d–g and Supplementary Fig. 4a, Supplementary Movie 5). Therefore, the priming of sheath assembly is also modulated by TagA rather than TssA alone[6]. In addition, deleting the TssA ImpA_N domain alone reduced sheath-tube levels similar to that of the Δ*tssA* Δ*tagA* mutant in contractile cells, and introducing the TagA ImpA_N deletion to the TssA ImpA_N deletion mutant did not lead to any further increase (Fig. 5b, e and Supplementary Fig. 4a, Supplementary

Movie 6), suggesting that the TssA ImpA_N deletion relieved TagA-mediated inhibition.

Since overexpression of TssM caused LPPS formation in both contractile and non-contractile cells (Fig. 2a, d), we tested its effect in the Δ*tssA* mutant. Unlike in TssM[+] and Δ*tssM* cells (Fig. 2d), overexpression of TssM did not lead to LPPS structures in Δ*tssA* mutant but substantially stimulated wild-type-like sheath-tube structures in both contractile and non-contractile cells (Fig. 5c–g and Supplementary Fig. 4c, Supplementary Movie 7). Although expression of TssM in the Δ*tssA* restored sheath-tube assembly to

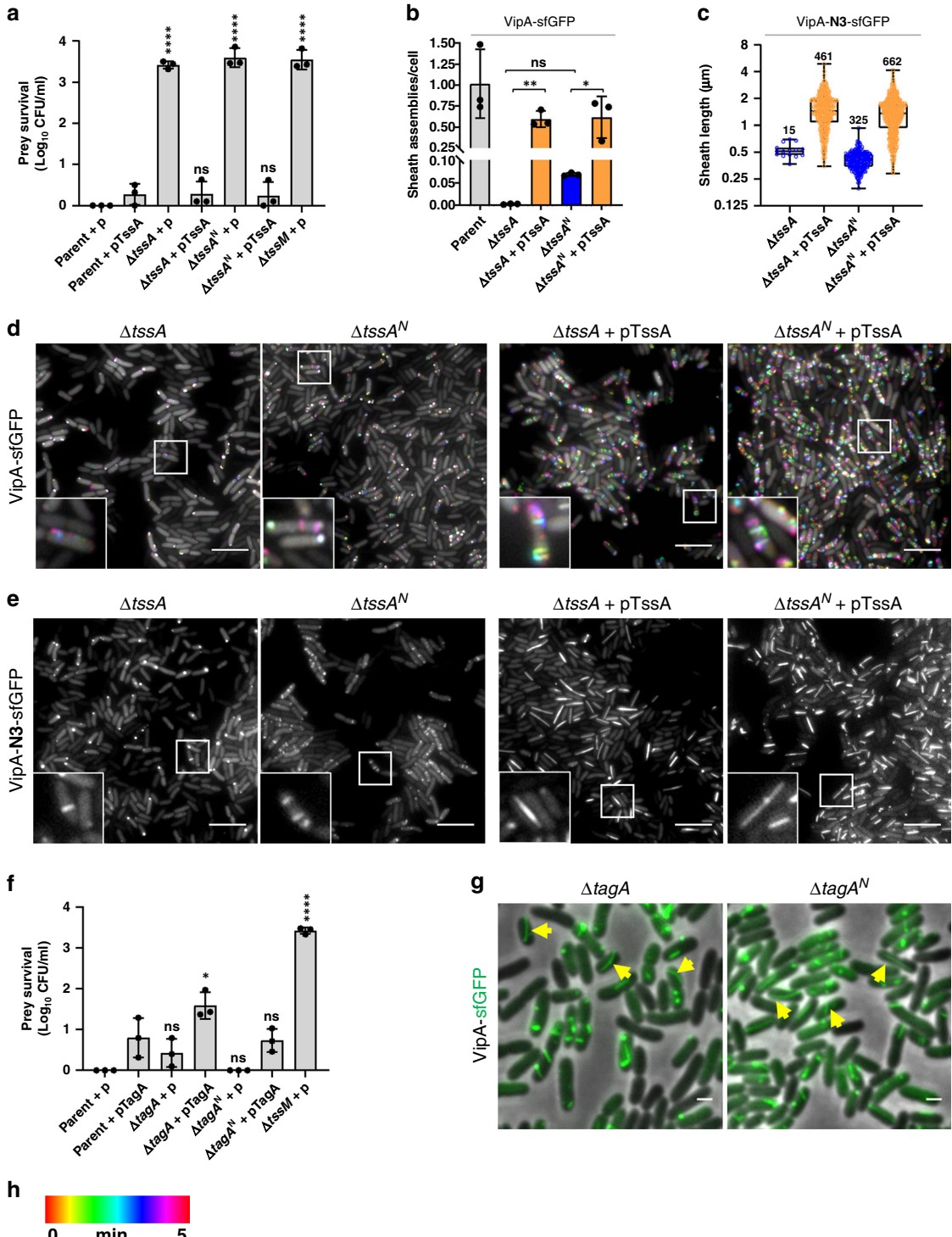

~15% of wild-type levels in contractile cells, it did not restore the killing of *E. coli* prey (Fig. 5e, Supplementary Fig. 4b). Cumulatively, these data suggest that TagA inhibits sheath assembly in the absence of TssA, and overexpression of TssM may sequester TagA to alleviate this inhibition.

~15% of wild-type levels in contractile cells, it did not restore the killing of *E. coli* prey (Fig. 5e, Supplementary Fig. 4b). Cumulatively, these data suggest that TagA inhibits sheath assembly in the absence of TssA, and overexpression of TssM may sequester TagA to alleviate this inhibition.

**TagA interacts with multiple T6SS structural proteins.** Using pull-down assays, we next examined the interaction of TssA and TagA with all T6SS structural components except for the outer-membrane TssJ protein and one of the baseplate components TssG due to poor expression in *E. coli*. We also compared our

findings with the known interactions previously reported in *P. aeruginosa*[31], enteroaggregative *E. coli*[28,30] and *V. cholerae*[29] (Fig. 6a). In addition to confirming known interactions of TssA and TagA with the sheath-tube and MC-BP proteins, we also detected a number of previously unknown interactions of TagA with Hcp and VgrG proteins, as well as with MC-BP proteins TssEF and TssLM (Fig. 6a and Supplementary Fig. 5a–g). We also found that both the N-terminus and the C-terminus of TssA could interact with TagA full-length protein and the N-terminus of TagA (Supplementary Fig. 5h). Interaction between the C-terminal fragments of TssA and TagA was also detected (Supplementary Fig. 5i). These results suggest that TagA also acts as a

**Fig. 4 ImpA_N domain of TssA and TagA is dispensable for T6SS initiation. a** Relative survival of *E. coli* prey after co-incubation with parental, complete deletion (Δ*tssA*) or ImpA_N domain deletion (Δ*tssA^N*) of *tssA* killer cells carrying pBAD plasmid, empty (p) or expressing *tssA* (pTssA); $n = 3$ independent experiments. **b** Sheath assemblies during 5 min time-lapse microscopy (assembled sheaths over total cells) in parental, Δ*tssA*, Δ*tssA^N* and complemented strains (pTssA) in the contractile VipA-sfGFP cells. One-way ANOVA with Sidak's multiple comparisons test comparing deleted to complemented strains and complete deletion to domain deletion strains *$p < 0.05$, **$p < 0.01$, ns, not significant. $n = 3$ independent experiments. **c** Sheath length (in μm) of Δ*tssA* and Δ*tssA^N* strains with or without overexpression of pTssA in the non-contractile VipA-N3-sfGFP. *Y* axis shows Log2 scale. Two-tailed Mann–Whitney test comparing each strain with and without overexpression of TssA was performed ($p < 0.001$). Whiskers show Min to Max values and boxes indicate median and extend from the 25th to 75th percentiles. Each circle represents a sheath, *n* is indicated in the graph. Data were acquired from three independent experiments. **d** Temporal color-coded images (30 ×30 μm) show sheath assembly in Δ*tssA* and ImpA_N domain deletion (Δ*tssA^N*) strains and complemented (pTssA) strains in VipA-sfGFP contractile background. **e** Non-contractile VipA-N3-sfGFP *tssA* deletion and ImpA_N *tssA* domain deletion strains and the same strains overexpressing pTssA. **f** Relative survival of *E. coli* prey after co-incubation with parental, *tagA* or ImpA_N domain *tagA* killer cells carrying pBAD plasmid, empty (p) or expressing *tagA* (pTagA); each dot represents an independent experiment ($n = 3$). **g** Images of complete deletion (Δ*tagA*) or ImpA_N domain deletion (Δ*tagA^N*) in VipA-sfGFP contractile cells. Images are a merge of phase and green fluorescent channel. Scale bar, 1 μm. Yellow arrows indicate curved sheaths. **h** Color-coded (spectrum) time scale corresponds to images in (**d**). In **a**, **f** One-way ANOVA with Sidak's multiple comparisons test comparing each deletion strain to the parental strain carrying the same plasmid, *$p < 0.05$, ****$p < 0.0001$, ns, not significant. In **a**, **b**, **f** error bars indicate the mean value ± SD of three biological replicates. In **d**, **e** insets (3.2 × 3.2 μm) correspond to area in white box. Scale bars, 5 μm.

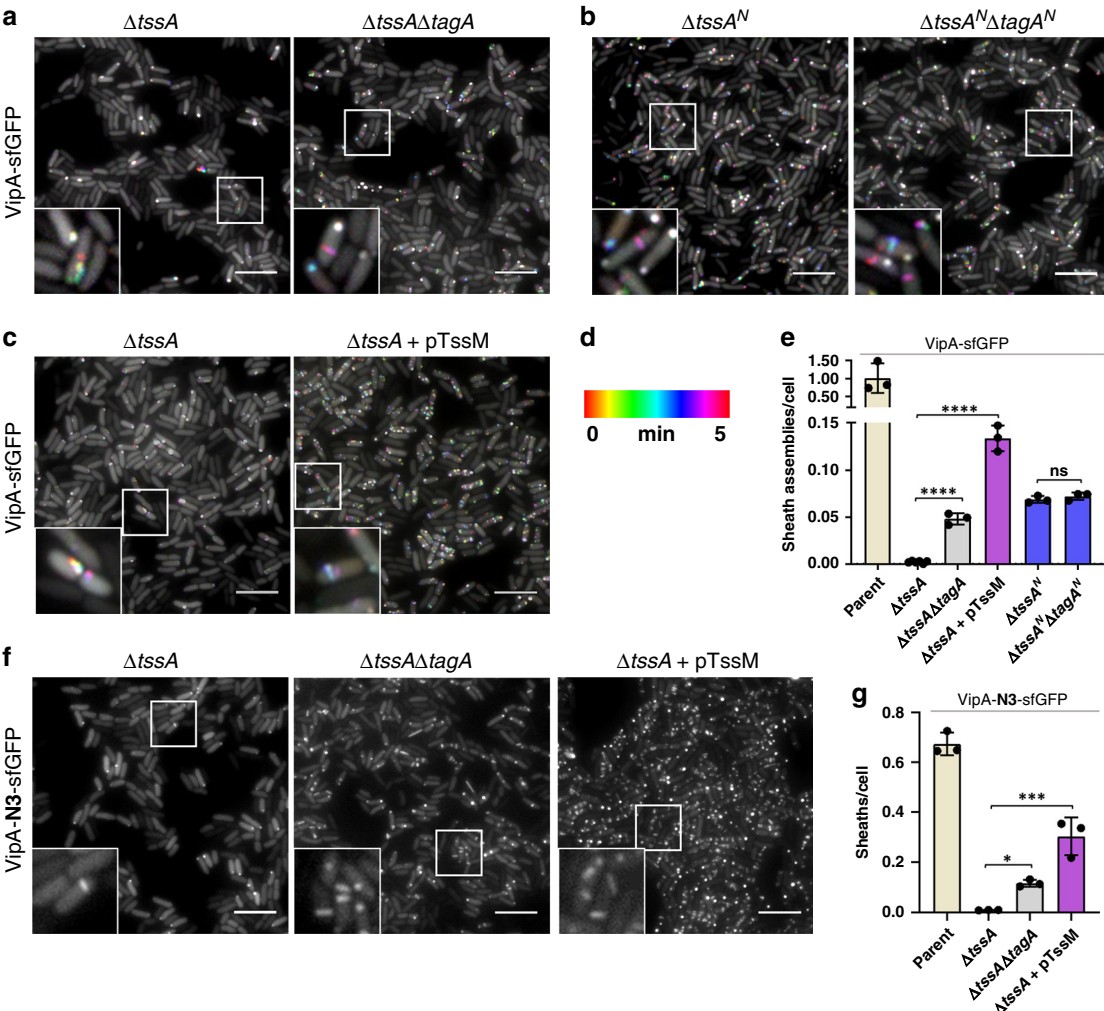

**Fig. 5 Deletion of *tagA* and induction of TssM stimulates sheath assembly in the absence of TssA.** Temporal color-coded images (30 ×30 μm) of contractile VipA-sfGFP *tssA* deletion (Δ*tssA*) only or double *tssA* and *tagA* deletion (Δ*tssA*Δ*tagA*) (**a**); ImpA_N domain deletion of TssA only (Δ*tssA^N*) or double domain deletion (Δ*tssA^N*Δ*tagA^N*) (**b**); and Δ*tssA* strain without (left) or with (right) expression of pTssM plasmid (**c**). **d** Color-coded (spectrum) time scale corresponds to images (**a**–**c**). **e** Bar graph shows sheath assemblies during 5 min time-lapse videos over total cells for parental, single *tssA* deletion, double *tssA* and *tagA* deletion, Δ*tssA* expressing *tssM* (pTssM), and ImpA_N domain deletion (Δ*tssA^N*), single and double domain deletion strains. $n = 3$ independent experiments. **f** Still images (30 ×30 μm) of non-contractile VipA-N3-sfGFP cells of single *tssA* and double *tssAtagA* deletion and Δ*tssA* strain expressing pTssM. **g** Graph shows the number of non-contractile sheaths over total cells in strains listed in (**f**); $n = 3$ independent experiments. All scale bars, 5 μm. Insets (3.2 ×3.2 μm) in all images correspond to area in white box. *$p < 0.05$, ***$p < 0.001$, ****$p < 0.0001$, ns, not significant, one-way ANOVA with Sidak's multiple comparison test. Each data point represents an independent biological replicate. Error bars indicate the mean value ± SD of three biological replicates.

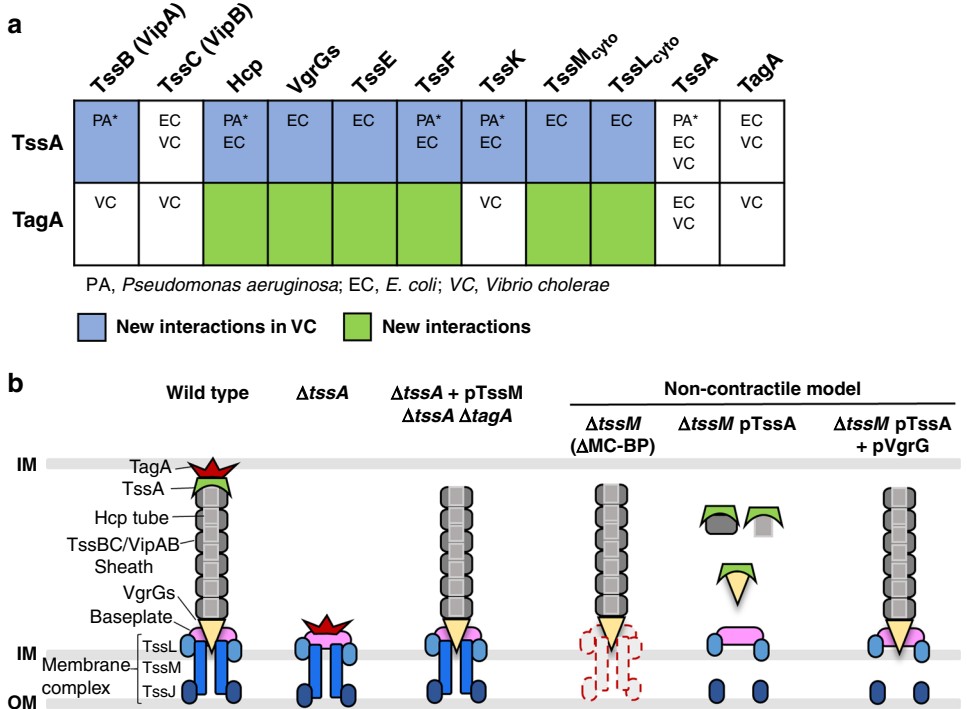

**Fig. 6 TssA–TagA–TssM regulates priming and termination of the T6SS assembly. a** Novel interaction partners of TssA and TagA in *Vibrio cholerae*. Table summarizes all tested T6SS proteins (columns) interacting with His-tagged TssA or TagA (rows) identified in this study. Protein interactions that were previously identified in other studies are indicated in each box (PA, *P. aeruginosa* by Planamente et al.[31]; EC, Enteroaggregative *E. coli* by Zoued et al.[28] and Santin et al.[30]; VC, *V. cholerae* by Schneider et al.[29]). Note that TssA in VC and EC belong to the TsaC functional class, while TssA1 in PA belongs to the TsaB functional class, thus denoted with asterisk. TagA in VC and EC belong to the TsmA class. Blue boxes indicate TssA protein interactions not previously shown in *V. cholerae*. Green boxes indicate TagA new interactions. All pull-down assays are included in Supplementary Fig. 5. **b** Schematic summarizing TssA–TssM–TagA role in T6SS dynamics. Current model, depicted as wild type, states that the sheath-tube structure is connected to a MC-BP complex at the proximal end and terminated by a TssA–TagA complex at the distal end. Priming of polymerization is controlled by TssA. Our results support that in the Δ*tssA* mutant, TagA inhibits initiation by interacting with MC-BP, while deletion of *tagA* and expression of TssM could relieve such inhibition. We further show that non-contractile sheath-tube could be assembled in the Δ*tssM* mutant, and other MC-BP deletion mutants. Assembly in the Δ*tssM* mutant could be inhibited by TssA expression and derepressed by co-expression of VgrG with TssA. These results highlight the priming and termination of T6SS assembly is regulated by TssA–TagA–TssM.

promiscuous T6SS chaperone in addition to its known role as a sheath-extension stopper.

## Discussion

Dissecting the functions of T6SS MC-BP components has been challenging since deleting any of them leads to the same phenotype, an impaired T6SS. As a result, the MC-BP are generally viewed as structural building blocks while what each component does during the assembly process has not been explored in detail. Here, we used the non-contractile T6SS sheath model and a panel of MC-BP mutants to differentiate how each MC-BP protein contributes to initiation and stability of sheath-tube polymerization. We report a number of conditions that produce long pole-to-pole sheath-tube structures, termed LPPS, in the absence of an intact MC-BP. In particular, we were able to distinguish the two inner membrane MC proteins TssM and TssL with distinct phenotypes in controlling sheath polymerization and protomer aggregation, respectively (Fig. 1f). Although the LPPS structures can be observed in all MC-BP mutants, the Δ*tssM* mutant showed the most abundant, at least 10 times more frequent than the others (Fig. 1b). These results demonstrate that an intact MC-BP is not essential for initiating the sheath-tube assembly but is key for maintaining the pre-contraction stability of a polymerizing contractile sheath-tube.

How is the LPPS generated? We noticed that sheath-tube structures, while ultimately becoming LPPSs, appear to originate randomly throughout the cell, rather than being limited to the pole area (Supplementary Fig. 1g, h and Supplementary Movie 1). We postulate that the observed LPPS structures in the Δ*tssM* cells is due to impaired termination and/or anchoring. Long and sometimes curved sheath-tube structures have been previously reported when the polymerizing sheath-tube cannot be terminated in the Δ*tagA* mutant[11,29,30,32,33]. Consistent with a recent report[29], we also found that the membrane-bound initiating point of a straight and growing T6SS contractile sheath-tube structure could be relocated to a different site in the Δ*tagA* mutant, after which the structure continued to extend for at least 10 s prior to contraction (Supplementary Fig. 3d). In this case, BP seems to be disconnected from the original MC and possibly reconnected with a new MC, allowing the sheath-tube to bypass the spatial limitation of the opposite membrane to continue growth regardless of the original orientation and eventually grow from pole-to-pole. However, it remains elusive how LPPS structures could initiate in defective MC-BP mutants. Because LPPS is most abundant in the Δ*tssM* mutant while mutants lacking individual BP components exhibit reduced formation of LPPS (Fig. 1b), the theoretically intact BP in the Δ*tssM* may initiate sheath-tube polymerization independent of the MC at higher efficiency than in the individual BP mutants. Future work on visualization of LPPS using Cryo-Electron Tomography would provide valuable evidence on assembly in those MC-BP mutants.

Sheath-tube termination is known to be controlled by TagA[29,30]. Here, our data clearly indicate that TssM and TssA act together

with TagA to control termination, which is evidenced by the formation of LPPS in the $\Delta tssM$ mutant or in TssM-induced cells (Fig. 2a, c, d) and the conflicting effect of TssA on LPPS formation in TssM$^+$ and $\Delta tssM$ cells (Fig. 3a, c). Considering the known multi-component interacting capability of TssA (Fig. 6a)[28–31] and its proposed role as a key chaperone[29], formation of LPPS when TssA or TssM is induced might result from sequestration of TagA by TssA or TssM in excess. Similarly, TssA induction inhibits LPPS in the $\Delta tssM$ mutant as well as sheath aggregation in the other BP-MC mutants but co-expression of VgrG suppresses such inhibition (Fig. 3d, e and Supplementary Fig. 2a, b). These data suggest that excess TssA might sequester VgrG or sheath-tube protomers, while a functional MC-BP is able to disassociate TssA-bound protomers for polymerization.

Another key observation we have made in this study is that expression of TssM in the $\Delta tssA$ mutant leads to frequent wild-type-like sheath formation but not LPPS (Fig. 5c–g). This result not only reveals that TssA is not required for initiating the T6SS assembly but also that the previously suggested TssA–TagA contact at the distal end[30] is not required for termination. Importantly, *tagA* deletion in the $\Delta tssA$ mutant could also substantially increase the number of sheath-tube structures (Fig. 5a, b, e). Because TssM interacts with TagA (Fig. 6a, Supplementary Fig. 5f), overexpression of TssM might interfere with TagA-mediated T6SS inhibition. Interestingly, about 10% of TagA was recently reported to be localized to sheath assembly initiation sites prior to assembly and 10% of TagA was found to be associated at the distal end of contracted sheath[29]. In addition, the number of sheath-tube assembled is higher in the $\Delta tssA$ mutant overexpressing TssM than the $\Delta tssA\Delta tagA$ mutant (Fig. 5e, g), suggesting that TssM plays an additional role in facilitating T6SS assembly that complements TssA deficiency. Although the mechanism is still unknown, the role of TssA is clearly more complex than previously considered as a priming factor.

To reconcile these abovementioned complex phenotypes, we propose a revised model (Fig. 6b) in which we place TssA–TssM–TagA at the center of controlling T6SS initiation and termination. We consider TssA–TagA as a pair of antagonistic chaperones that interact with multiple T6SS MC-BP and sheath-tube components, including the VgrG spike. TssA could facilitate T6SS polymerization by either freely shuttling sheath-tube protomer to the polymerizing site or remain attached to the extending end to recruit protomers. Its role is antagonized by its interacting partner TagA, which not only could terminate sheath-tube growth at the distal end but also inhibit initiation by binding to the MC-BP and VgrG spike components. In addition, a fully functional MC-BP, or TssM directly, can disassociate TssA-VgrG or other TssA-bound structural components to facilitate their loading onto the sheath-tube structure, while TssL could prevent abnormal aggregation of sheath-tube protomers. Lastly, BP and MC are functionally modular as supported by the observed translocation of an initiating BP to another MC while extending (Supplementary Fig. 3d and ref. [29]) and the abundant LPPS formation in the MC defective $\Delta tssM$ mutant (Fig. 2d). Collectively, our results highlight the complexity of T6SS initiation and termination mediated by the TssA–TssM–TagA trio and the remarkable evolutionary process that integrates the T4SS-orginated MC with the phage-originated BP-sheath, making the hybrid T6SS a powerful weapon for bacterial interspecies interactions.

## Methods

**Bacterial strains and growth conditions**. Bacteria were grown aerobically in LB media (1% [wt/vol] tryptone, 0.5% [wt/vol] yeast extract, 0.5% [wt/vol] NaCl) at 37 °C with shaking at 200 r.p.m. A complete list of strains and plasmids used in

this study can be found in Supplementary Tables 1 and 2, respectively. The suicidal vector pDS132 was used for chromosomal recombination to generate gene knockout strains in *V. cholerae*[36]. The non-contractile sheath phenotype in all strains was generated by using the pDS132-VipA-N3-sfGFP plasmid that carries a 3 amino acid insertion in the N-terminal domain of VipA[26]. Expression plasmids were created using standard PCR and restriction enzyme digest protocols. All vectors and strains were confirmed by sequencing. Antibiotics were added to overnight cultures at the following concentrations: 50 μg/mL kanamycin, 75 μg/mL carbenicillin, 20 μg/mL gentamycin, chloramphenicol 25 μg/mL (for *E. coli*) and 2.5 μg/mL (for *V. cholerae*). L-arabinose was added to the growth media at the concentrations specified below for each assay.

**Bacterial killing assay**. Bacterial cultures of killer cells grown overnight (~15 h) were diluted 1:100 in fresh LB media and grown to $OD_{600}$ = 0.8–1.0. To induce expression of pBAD plasmid, 0.01% [wt/vol] L-arabinose was added to the liquid media for 15 min. Cells were then harvested and mixed with overnight grown *E. coli* MG1655 prey at either 10:1 or 20:1 (only for assays depicted in Supplementary Fig. 4a, b) ratio (killer: prey). The mix was spotted onto an LB + 0.01% [wt/vol] L-arabinose plate and incubated for 3 h at 37 °C. Cells were then collected and resuspended in 500 μl of LB and 10-fold serial dilutions were plated onto LB agar with 20 μg/mL gentamycin to assess survival of *E. coli* MG1655 prey. All killing assays were performed in triplicate. The mean $\log_{10}$ colony-forming units (CFU) of prey survival in three biological replicates is plotted in bar graphs and the standard deviation (±SD) is indicated with error bars.

**Protein pull-down assays**. His-tagged bait and V5-tagged prey genes were cloned into pETDuet and pBAD vectors. His-tagged TssA$^N$ (E2-Q126) was prone to precipitation when expressed, thus a His-tagged maltose binding protein (MBP)-TssA$^N$ with increased solubility was used instead. *E. coli* BL21 DE3 carrying different pETDuet plasmids or *E. coli* T-Fast with different pBAD plasmids were grown in 10 ml LB media (1% [wt/vol] tryptone, 0.5% [wt/vol] yeast extract, 0.5% [wt/vol] NaCl) with appropriate antibiotic at 37 °C to an $OD_{600}$ of approximately 0.6. Cells were induced with 1 mM IPTG at 20 °C for 16 h or with 0.1% [wt/vol] L-arabinose at 30 °C for 3 h. Cells were then harvested and resuspended in 1 ml of lysis buffer (20 mM Tris (pH 8.0), 500 mM NaCl and 50 mM imidazole with 1× Halt protease inhibitor cocktail (Thermo Scientific)). After sonication, the lysates were collected by centrifugation (13,800 × g for 10 min). Lysate and Ni-NTA resin (Smart lifesciences) were mixed and incubated at 4 °C for 1 h, washed with 1 ml wash buffer (20 mM Tris (pH 8.0), 500 mM NaCl and 50 mM imidazole), eluted in 100 μl elution buffer (20 mM Tris (pH 8.0), 500 mM NaCl and 500 mM imidazole) and analyzed by western blot.

**Western blot analysis**. Protein samples (10 μl) were run on a 12% SDS–PAGE gel and transferred to a polyvinylidene fluoride (PVDF) membrane (Bio-Rad) by electrophoresis. Then, membranes were blocked with 5% non-fat milk in TBST buffer (50 mM Tris, 150 mM NaCl and 0.05% Tween-20, pH 7.6) for 1 h at room temperature, and incubated with primary antibody (1:10,000 dilution, mouse monoclonal antibody anti-6× HIS from Sigma Aldrich and anti-V5 from Thermo Scientific) in TBST with 1% non-fat milk for 1 h at room temperature. The membranes were washed three times for 10 min in TBST buffer, and incubated with an anti-mouse IgG horseradish peroxidase (HRP)-conjugated secondary antibody (Cell Signaling Technology) in TBST with 1% non-fat milk for 1 h. After washing three times for 10 min with TBST buffer, we detected the signals using the ECL Clarity solution (Bio-Rad) and a Chemi Scope system (Clinx Chemi Scope 3300 Mini). Uncropped blots are included in the Source data file.

**Sample preparation for fluorescence microscopy**. Bacterial cultures were grown overnight for 15 h, diluted 1:100 into fresh LB media and incubated in shaker at 37 °C until reaching $OD_{600}$ = 0.8–1.0 (~2.5 h). When indicated, 0.1% [wt/vol] (for strains with contractile T6SS sheath) or 0.4% [wt/vol] (for strains carrying the N3 non-contractile mutation) L-arabinose was added to the liquid media for induction of pBAD plasmids for 2.5 h for non-contractile VipA-N3 strains or 30 min for the contractile VipA intact strains. The cultures were then centrifuged for 1 min at 13,800 × g, concentrated to $OD_{600}$ = 20 and resuspended in 0.5× PBS. 1 μL aliquot of bacteria was spotted onto 1% agarose-0.5× PBS pads and imaged immediately.

**Image acquisition**. Time-lapse widefield fluorescence microscopy images were acquired with a Nikon Ti-E inverted microscope containing a CFI Plan Apochromat Lambda ×100 oil objective and using the Perfect Focus System (PFS). The GFP fluorescent signal was excited and filtered using Intensilight C-HGFIE (Nikon) and ET-GFP (Chroma 49002) filter set. ANDOR Clara camera (DR 328G-C01-SIL) was used to record the images with a pixel size of 60 nm and the NIS-Elements AR 4.40 software to visualize the videos. All images included in this study correspond to time-lapse videos acquired at a frame rate of 10 s per frame. All strains with contractile sheaths (VipA-sfGFP) were imaged during 5 min while the

less dynamic strains carrying the non-contractile mutation VipA-N3 were imaged for 3 min.

**Image analysis**. All images were processed using Fiji[37]. Time-lapse videos were corrected for photobleaching by normalizing the fluorescence intensity to the same mean intensity for each image in a time series as described previously[38]. The StackReg plugin for Fiji was used to align all images in a stack. Images were taken in at least three independent biological replicates. 40 ×40 μm fields of views were cropped from each image for analysis. To analyze the T6SS dynamics in the strains with intact VipA-sfGFP, the Temporal Color Code plugin for Fiji was used with the LUT set to spectrum. A time color scale bar is shown in the figures and described in figure legends. For non-dynamic strains carrying the VipA-N3 mutation, only one frame was used to count and measure T6SS sheaths. LPPS structures were defined as sheaths extending pole-to-pole along the length of the cell and, in some cases, exciding cell length and bending. The length of the sheaths was measured using the straight-line tool for Fiji. For bent or curved sheaths, the freehand line tool was used instead. Foci were manually counted and recorded. The total number of cells was assessed on the brightfield channel by using the Analyze Particles function of Fiji and manually validated.

**Quantification and statistical analysis**. One-way ANOVA with Sidak's multiple comparisons or Two-tailed Mann–Whitney test was performed using GraphPad Prism version 8.0.0. Number of cells analyzed ($n$) as well as significance of each comparison are indicated in the figures and figure legends. Unless indicated differently, values are expressed as mean ± SD. All experiments were performed in at least three independent replicates. The exact $p$ values for all statistical analyses are provided in the Source data file.

**Reporting summary**. Further information on research design is available in the Nature Research Reporting Summary linked to this article.

## Data availability
Data supporting the current study are available from the corresponding author on request. Source data are provided with this paper.

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

## Acknowledgements
This work was supported by grants from National Natural Science Foundation of China (31770082), the Canadian Institutes of Health Research and the Natural Sciences and Engineering Research Council (NSERC) to T.G.D. T.G.D. was supported by Canada Research Chair program.

## Author contributions
M.S.S. performed experiments, analyzed results and prepared figures and manuscript, X.L., H.L., and M.S.S. constructed plasmids and strains, X.Z. performed bacterial killing assays, H.L. performed protein interaction assays, M.S.S and T.G.D wrote the paper, T.G.D. conceived and supervised the project.

## Competing interests
The authors declare no competing interests.
