## [Peer Review File · Nature Communications]

Reviewers' comments:

Reviewer #1 (Remarks to the Author):

The manuscript submitted by Stietz et al. describes experiments that were designed to shed light on priming and assembly of the T6SS sheath. They largely involve comparing the effect of expressing the contractile competent and a contractile incompetent variant of the sheath protein VipA (TssB) in various *V. cholerae* T6SS mutants or following T6SS subunit overexpression, followed by analysis using an unspecified type of fluorescence microscopy. Some interaction studies involving components of the T6SS were also performed.

I found the first part of the Results section difficult to follow due to insufficient detail. For example, in the first Results subsection (Page 5, L6), the authors refer to '...the non-contractile VipA-N3-sfGFP construct...' There are no details provided for this mutant anywhere in the paper. It is fundamental to an understanding of the work. I found a reference to the construct (Stietz et al., 2019) in the Extended Data Table to which I referred. This states 'the vipA-N3 mutant, in which the sheath-Hcp tube is locked into the extended form' and refers the reader to Brackmann et al., 2017. Here I learn that N3 is a 3 aa insertion in the N-terminal linker of TssB (VipA). The authors should insert a sentence either in the Introduction or in this section of the paper explaining to the reader the nature of the mutant.

The authors then describe what they observe in T6SS mutant cells expressing GFP-tagged WT or mutant VipA subunits (results depicted in Figure 1 and S1). The authors state that polar foci are observed in some T6SS mutants but there is no indication as to what this implies. It means that VipA-GFP is associating with a pole of the cell, but is it forming normal sheaths, short sheaths, just one 'layer' etc? Some indication as to what the norm is would be helpful to the reader. This is compounded by the fact that a wild-type strain was not included as a comparator in the figures, which I found to be a strange omission. Based on the way the first subsection is written, I would conclude that the formation of polar foci corresponds to normal sheath formation. However, my conclusion may be erroneous. For example, in the first Results subsection it is stated: 'mutants of the BP gene Δ tssK and the MC genes Δ tssJ and Δ tssM showed more polar foci'. I assume this to mean that tssM is not required for contractile sheath formation. However, in the second Results subsection, it is stated: 'Deletion of tssM expectedly abolished assembly of the contractile sheath.'

The interaction studies are not completely novel, as the Cascales group and others have reported separate analyses on the interaction of TssA with other subunits. To my knowledge, interaction studies between TagA and other subunits have not been previously reported, and the results are not consistent with the current model for TagA function as reported by Santin et al., 2018. My assessment of the TagA-TssA interaction results do not concur with those of the authors. For example, it is stated that the TssA C-terminal region directly interacts with the C- and N-terminal domains of TagA, but the pulldown result with the TagA C-terminal region as prey (Figure 6B) is not wholly convincing. Surely, the result suggests that TagA and the TagA N-terminal region interact with the N- and C-terminal regions of TssA? The authors also conclude that the N-terminal ImpA_N domain of TssA interacts with the C-terminal domain of TagA but not with its N-terminal domain. My interpretation of Figure 6B, is that the N-terminal domain of TagA, not the C-terminal domain, is being pulled down by the TssA N-terminal region.

The authors should revise the TssA and TagA nomenclature to be consistent with the proposal of the Basler group, to which they refer in the Introduction. There are at least three different members of the TssA family based on their completely unrelated C-terminal regions, and the different members also appear to have different functions, thereby giving rise to the new nomenclature (TsaB, TsaC and TsaA). Moreover, use of the designation TagA rather than TsaA can lead to confusion, as TagA was assigned to a different Tag protein by Shalom et al., 2007.

It is very difficult to draw any firm conclusions regarding what the data indicate, and the results with the non-contractile mutant somewhat confuses the analysis. Also, I find it unlikely that TagA, a cytoplasmic membrane-associated protein, can behave as a cytoplasmic protein and 'chelate' VgrG as shown in Figure 7D.

Introduction:

P4, L3-4: The N-terminal domain (ImpA_N) does not form dimers. Zoued et al., 2016 showed that an N-terminal construct referred to as TssA Nt1 formed dimers. However, Nt1 contains the N-terminal ImpA_N domain and the middle domain that dimerises.

L10: not correct to refer to TagA as a TssA homologue. They both share a homologous NTD.

L17: ImpA_N refers to the conserved N-terminal domain, so it is not appropriate to use it here. Perhaps ImpA_N family or state TsaC and TsmA.

L18-19: '...the VgrG spike and the inner Hcp needle are expectedly required for assembly...' of what?

Results:

P5, L4: Systematic, not systemic

L13: 'In contractile-sheath cells,' meaning?

P7, L21: state what is meant by tssAN and tssAC as you did for Δ tssAN.

P9, L4: protomer

Reviewer #2 (Remarks to the Author):

In this work, the authors systematically examined the assembly of contractile and non-contractile sheaths in MC-BP mutants. They found stable non-contractile sheath-needle structures in each MC-BP mutant and the tssA mutant, suggesting that priming of T6SS is independent of an intact MC-BP and the priming factor TssA. Furthermore, the authors propose a revised model in which the intact MC-BP is required for stabilizing contractile structures but not for initiating polymerization. It is an interestingly work, however, the manuscript is very hard to follow.

1. The abstract does not make justice to the made work. Among other things, a nice conclusion at the end of the abstract is needed.
2. Some parts of the Introduction are confused; i.e. if you are using TssX nomenclature do not mix with the former VipAB. It also confused the description about the ImpA_N protein family. More explanation about "aberrantly long pole to pole sheath (ALPPS)" phenotype is also needed.
3. The Results section is the worst part. The subtitles should state the conclusion of that section, instead of the unspecific subtitles used (i.e. Systemic examination of sheath formation in MC-BP mutants).
4. VipA-N3-sfGFP sheath mutant experiments must be better explained (it is not clear in the M&M section too) including the reasoning of its use.
5. Some controls are needed; i.e. in Figure 1 A&B (i.e. what happen with the non-contractile VipA-N3-sfGFP construct without the mutations, Δ tssE, Δ tssF and Δ tssG, etc.).
6. In some subtitles in the Results section there is not a conclusion at the end.
7. The whole narrative of the Results section must be improved (i.e most of the information to

improve this section is written in the Discussion section). The titles in the figure legends are better than the subtitles in the Results section.

8. The Discussion section is not well written, since this information must be interesting to know before, in the Result section. The info in the Discussion section are explanations of the results instead of validation with the literature; thereby, only 5 references are included in the Discussion section ([27], [29,36], [29-32], [30] and [31]).

9. Only when the reader reaches the Figure 7, the whole story becomes clear. Thereby, the manuscript must be rewritten.

10. The M&M section lacks important details.

Response to Reviewers

Assembling a needle without an anchor: functional interaction of TssA-TssM-TagA dictates biogenesis and termination of the type VI secretion system in *Vibrio cholerae*

Stietz et al.

Point-by-point response (our response in blue)

We would like to thank both reviewers for the constructive comments. We took all suggestions to heart and have revised the manuscript throughout especially the results and discussion to make it more focused on the main findings. We believe these changes have improved the clarity and hope it is now easier for the readers to follow.

Reviewer #1 (Remarks to the Author):

The manuscript submitted by Stietz et al. describes experiments that were designed to shed light on priming and assembly of the T6SS sheath. They largely involve comparing the effect of expressing the contractile competent and a contractile incompetent variant of the sheath protein VipA (TssB) in various *V. cholerae* T6SS mutants or following T6SS subunit overexpression, followed by analysis using an unspecified type of fluorescence microscopy. Some interaction studies involving components of the T6SS were also performed.

I found the first part of the Results section difficult to follow due to insufficient detail. For example, in the first Results subsection (Page 5, L6), the authors refer to ‘...the non-contractile VipA-N3-sfGFP construct...’ There are no details provided for this mutant anywhere in the paper. It is fundamental to an understanding of the work. I found a reference to the construct (Stietz et al., 2019) in the Extended Data Table to which I referred. This states ‘the vipA-N3 mutant, in which the sheath-Hcp tube is locked into the extended form’ and refers the reader to Brackmann et al., 2017. Here I learn that N3 is a 3 aa insertion in the N-terminal linker of TssB (VipA). The authors should insert a sentence either in the Introduction or in this section of the paper explaining to the reader the nature of the mutant.

Thank you for the constructive comments. We have added more experimental details and references wherever appropriate throughout the manuscript. Specifically, for the VipA-N3 mutation, we have now introduced it in the Introduction P3 L23.

“...insertions in the VipA N-terminal linker region can block contraction and result in non-contractile sheath-tube structures locked in the pre-contraction state^{20,26}” as well as in Results P5 L15 “...we employed a previously established non-contractile sheath model with a 3 amino acid (Ala-Glu-Val) insertion in the VipA(TssB) N-terminal linker region²⁶ and chromosomally introduced this construct, hereafter referred to as VipA-N3-sfGFP, to a panel of individual MC-BP mutants.”

We have also added more details to microscopy analysis in the Methods P14-15 section.

In addition, we have also included a T6SS model in Fig1a to help readers understand protein functions.

a
The authors then describe what they observe in T6SS mutant cells expressing GFP-tagged WT or mutant VipA subunits (results depicted in Figure 1 and S1). The authors state that polar foci are observed in some T6SS mutants but there is no indication as to what this implies. It means that VipA-GFP is associating with a pole of the cell, but is it forming normal sheaths, short sheaths, just one ‘layer’ etc? Some indication as to what the norm is would be helpful to the reader. This is compounded by the fact that a wild-type strain was not included as a comparator in the figures, which I found to be a strange omission. Based on the way the first subsection is written, I would conclude that the formation of polar foci corresponds to normal sheath formation. However, my conclusion may be erroneous. For example, in the first Results subsection it is stated: ‘mutants of the BP gene Δ tssK and the MC genes Δ tssJ and Δ tssM showed more polar foci’. I assume this to mean that tssM is not required for contractile sheath formation. However, in the second Results subsection, it is stated: ‘Deletion of tssM expectedly abolished assembly of the contractile sheath.’

Thank you for pointing out this confusion. These polar foci are likely sheath aggregates. We have now added some explanation regarding what those polar foci are in the Results P5 L24 “Similar to Δ vgrG1&3 mutant, BP mutants and the outer-membrane MC gene Δ tssJ mutant exhibited predominantly polar foci that resemble inclusion bodies of protein aggregates³⁵”. In addition, we have also added the wild-type sample for comparison in Figure 1 as requested and described the ‘norm’ phenotype for this strain in the manuscript (P5, L18). We have also quantified number of sheaths over total cell (Figure 1b, ‘Parent’ yellow bar) and sheath length (Figure 1c) for VipA-N3-sfGFP wild type strain.

d WT, VipA-N3-sfGFP

For the contractile phenotype, we have included wild type VipA-sfGFP strain in a temporal color-coded format as Supplemental Figure 1d and quantified foci formation in this strain (Supplemental Figure 1f, 'Parent' yellow bar).

d WT, VipA-sfGFP

The interaction studies are not completely novel, as the Cascales group and others have reported separate analyses on the interaction of TssA with other subunits. To my knowledge, interaction studies between TagA and other subunits have not been previously reported, and the results are not consistent with the current model for TagA function as reported by Santin et al., 2018. My assessment of the TagA-TssA interaction results do not concur with those of the authors. For example, it is stated that the TssA C-terminal region directly interacts with the C- and N-terminal domains of TagA, but the pulldown result with the TagA C-terminal region as prey (Figure 6B) is not wholly convincing. Surely, the result suggests that TagA and the TagA N-terminal region interact with the N- and C-terminal regions of TssA? The authors also conclude that the N-terminal ImpA_N domain of TssA interacts with the C-terminal domain of TagA but not with its N-terminal domain. My interpretation of Figure 6B, is that the N-terminal domain of TagA, not the C-terminal domain, is being pulled down by the TssA N-terminal region.

We agree that TssA is known to interact with multiple subunits of T6SS, as we have referenced in the manuscript. However, TagA interaction isn't well characterized. To clearly highlight our findings, we have now provided a summary as Figure 6a. Because TssA and TagA seem to functionally differ in different species, we have also added species where these interactions are reported in the summary table.

We also agree with the reviewer’s interpretation of the pull-down results and added additional data to confirm the interaction between the TssA and the TagA C-terminal fragments (Supplemental Fig. 5i). To clarify, we have revised the statement in Results P9 L20 “We also found that both the N-terminus and the C-terminus of TssA could interact with TagA full-length protein and the N-terminus of TagA (Supplemental Fig. 5h). Interaction between the C-terminal fragments of TssA and TagA was also detected (Supplemental Fig. 5i).”

The authors should revise the TssA and TagA nomenclature to be consistent with the proposal of the Basler group, to which they refer in the Introduction. There are at least three different members of the TssA family based on their completely unrelated C-terminal regions, and the different members also appear to have different functions, thereby giving rise to the new nomenclature (TsaB, TsaC and TsmA). Moreover, use of the designation TagA rather than TsmA can lead to confusion, as TagA was assigned to a different Tag protein by Shalom et al., 2007.

We thank the reviewer for the suggestion. However, we wish to keep the current nomenclature TssA and TagA since they are currently more accepted by the field and in the literature including the most recent comprehensive review on the T6SS published by the Basler group, in which TssA and TagA are used (Wang et al, 2019)

1. Wang, J., Brodmann, M. & Basler, M. Assembly and Subcellular Localization of Bacterial Type VI Secretion Systems. *Annu. Rev. Microbiol.* 73, 621–638 (2019).

It is very difficult to draw any firm conclusions regarding what the data indicate, and the results with the non-contractile mutant somewhat confuses the analysis. Also, I find it unlikely that

TagA, a cytoplasmic membrane-associated protein, can behave as a cytoplasmic protein and ‘chelate’ VgrG as shown in Figure 7D.

To clarify, we have now highlighted in the figure and the main text whether the noncontractile VipA-N3-sfGFP or the contractile VipA-sfGFP is used wherever appropriate. Although TagA is found to be localized to the proximity of membrane, it doesn’t have any N-terminal Sec or Tat signal sequence that indicates it is embedded into the inner membrane. We argue that binding to these multiple T6SS proteins could determine its distribution in the cells and some Hcp or sheath-interacting TagA might be present in the cytosol. We have also added some discussion by pointing a recent finding by the Balsler group P11 L17. “Interestingly, about 10% of TagA was recently reported to be localized to sheath assembly initiation sites prior to assembly and 10% of TagA was found to be associated at the distal end of contracted sheath ²⁹.”

Thus it remains an interesting question and what determines TagA localization is an ongoing investigation in my lab.

Finally, to highlight the main findings, we prepared a new and simpler model as Fig 6b to focus on the function of TssM-TssA-TagA trio on T6SS initiation and termination.

Introduction:

P4, L3-4: The N-terminal domain (ImpA_N) does not form dimers. Zoued et al., 2016 showed that an N-terminal construct referred to as TssA Nt1 formed dimers. However, Nt1 contains the N-terminal ImpA_N domain and the middle domain that dimerises.

Thank you and we have now deleted this statement.

L10: not correct to refer to TagA as a TssA homologue. They both share a homologous NTD.

We agree and have corrected this.

L17: ImpA_N refers to the conserved N-terminal domain, so it is not appropriate to use it here. Perhaps ImpA_N family or state TsaC and TsmA.

We have now replaced ImpA_N with TssA and TagA in the statement below. “

In this study, we aim to address this question by comparing sheath assembly of wild type contractile sheath and mutant non-contractile sheath using a series of MC-BP, TssA and TagA mutants in *Vibrio cholerae*.”

L18-19: ‘...the VgrG spike and the inner Hcp needle are expectedly required for assembly...’ of what?

We have rephrased this section and removed this sentence.

Results:

P5, L4: Systematic, not systemic

Agree and changed.

L13: ‘In contractile-sheath cells,’ meaning?

This section has been replaced to better explain the T6SS non-contractile sheath model versus the normal contractile sheath-tube structures.

P7, L21: state what is meant by tssAN and tssAC as you did for □tssAN.

We have now changed this section by moving the interaction data to supplemental and added a schematic figure (Supplemental Figure S5a) to indicate what TssA_N and TssA_C are.

P9, L4: protomer

Thank you and it has been corrected.

Reviewer #2 (Remarks to the Author):

In this work, the authors systematically examined the assembly of contractile and non-contractile sheaths in MC-BP mutants. They found stable non-contractile sheath-needle structures in each MC-BP mutant and the tssA mutant, suggesting that priming of T6SS is independent of an intact MC-BP and the priming factor TssA. Furthermore, the authors propose a revised model in which the intact MC-BP is required for stabilizing contractile structures but not for initiating polymerization. It is an interestingly work, however, the manuscript is very hard to follow.

Thank you for the comments and constructive suggestions.

1. The abstract does not make justice to the made work. Among other things, a nice conclusion at the end of the abstract is needed.

We have revised the abstract substantially and believe it conveys the message much better now.

2. Some parts of the Introduction are confused; i.e. if you are using TssX nomenclature do not mix with the former VipAB. It also confused the description about the ImpA_N protein family. More explanation about “aberrantly long pole to pole sheath (ALPPS)” phenotype is also needed.

Thank you for the suggestion. VipAB are commonly used in the field especially in *V. cholerae* T6SS papers. To clarify, we have indicated both nomenclature whenever needed, as exemplified in the introduction “VipAB (also known as TssBC)”. Similarly, we have also added more description to the ImpA_N family and the long pole to pole sheath (LPPS) in this revision in the Introduction and the beginning of Results.

3. The Results section is the worst part. The subtitles should state the conclusion of that section, instead of the unspecific subtitles used (i.e. Systemic examination of sheath formation in MC-BP mutants).

Thank you for the suggestion. We have revised all subtitles to state the main findings.

4. VipA-N3-sfGFP sheath mutant experiments must be better explained (it is not clear in the M&M section too) including the reasoning of its use.

We have added more details in the Introduction, Results and Methods regarding the construction of this VipA-N3-sfGFP, as stated in the response to reviewer 1. We have also included a rationale for using the non-contractile construct because it helps us to differentiate defects in initiation and stability of these T6SS structural gene mutants.

see P3 L23 and P5 L13

P3 L23 “but insertions in the VipA N-terminal linker region can block contraction and result in non-contractile sheath-tube structures locked in the pre-contraction state^{20,26}. ”

P5 L13 “To distinguish the contribution of each MC-BP component to sheath assembly and stability, we employed a previously established non-contractile sheath model with a 3 amino acid (Ala-Glu-Val) insertion in the VipA(TssB) N-terminal linker region²⁶ and chromosomally introduced this construct, hereafter referred to as VipA-N3-sfGFP, to a panel of individual MC-BP mutants.”

5. Some controls are needed; i.e. in Figure 1 A&B (i.e. what happen with the non-contractile VipA-N3-sfGFP construct without the mutations, Δ tssE, Δ tssF and Δ tssG, etc.).

This was raised by reviewer 1 too. We have now added wild type VipA-N3-sfGFP parental strain as Figure 1d for comparison and described the ‘norm’ phenotype for this strain in the manuscript (Page 5, L17-20). We have also quantified number of sheaths over total cell (Figure 1b, ‘Parent’ yellow bar) and sheath length (Figure 1c) for VipA-N3-sfGFP wild type strain. For the contractile VipA-sfGFP cells, we included wild type VipA-sfGFP strain in a temporal color-coded format as Supplemental Figure 1d and quantified foci formation in this strain (Supplemental Figure 1f, ‘Parent’ yellow bar). Please also see response to reviewer 1.

6. In some subtitles in the Results section there is not a conclusion at the end.

We have revised the subtitles and added concluding statement for each section.

7. The whole narrative of the Results section must be improved (i.e most of the information to improve this section is written in the Discussion section). The titles in the figure legends are better than the subtitles in the Results section.

We have revised the results and discussion to improve clarity.

8. The Discussion section is not well written, since this information must be interesting to know before, in the Result section. The info in the Discussion section are explanations of the results instead of validation with the literature; thereby, only 5 references are included in the Discussion section ([27], [29,36], [29-32], [30] and [31]).

We have revised the discussion substantially and hope it is much easier to follow now.

9. Only when the reader reaches the Figure 7, the whole story becomes clear. Thereby, the manuscript must be rewritten.

Thank you and we believe the revised manuscript has improved in clarity and logic follow in comparison with the first submission.

10. The M&M section lacks important details.

We have also revised this section as well to include more experimental details and hope these changes have addressed all the concerns.

REVIEWER COMMENTS

Reviewer #1 (Remarks to the Author):

Stietz et al analyse the consequences of deleting T6SS subunit genes or overexpressing them in wild type cells and cells encoding an aberrant VipA (TssB) sheath subunit that assembles into sheaths that do not contract. The results are somewhat surprising given current models for T6SS function, and in some cases difficult to explain without a degree of speculation. Based on the interpretation of their observations, the authors propose a revised model to account for the roles of TssM, TssA and TagA in assembly of the tube-sheath complex.

This version of the manuscript is certainly easier to follow than the previous one, although there are still places where things are not crystal clear (which I indicate below).

Title:

What does 'termination' refer to in this context? Surely "...biogenesis and termination of T6SS sheath-tube assembly in *Vibrio cholerae*" would be more accurate?

Abstract:

"Our results demonstrate that T6SS priming is not dependent on TssA, nor is its termination dependent on the distal end TssA-TagA interaction..."

Again, use of the term 'termination' needs to be more specific. The T6SS does not terminate. The authors mean termination of tube-sheath assembly.

Introduction:

P3, line 21 and P12, L6, the term 'sheath-needle' is not replaced by 'sheath-tube' as stated by the authors in their comment on P3, L11.

'Needle' is also used in isolation, i.e. Fig. 1 legend. This should be defined somewhere.

P3, L17-18, The authors state "The tip protein PAAR is critical in *Acinetobacter baylyi* ADP1 but dispensable in *Vibrio cholerae* for T6SS function" and yet it is clearly stated in the abstract to the reference cited by the authors (Shneider et al., 2013) that PAAR proteins are essential for the functioning of the *V. cholerae* T6SS.

P4, L6-10, confusing. In the first sentence, the authors refer to TssA proteins that serve as chaperones as TssA or TsaC, whereas TssA proteins that remain baseplate associated are referred to only as TsaB. This would be OK if the authors stick to this definition. However, in the second sentence, where they are referring to TssA (and therefore, as far as the reader is concerned, they are referring only to TsaC), the authors provide references for the interactions of both types of TssA (the Planamente et al paper deals with TsaB). Moreover, they mention interactions of TssA with TssC, E, F, G, L, K, M, Hcp, VgrG and TagA. What TssA are they referring to, their definition (TsaC) or the broader definition (TsaB and TsaC)? TsaC does not interact with all of these proteins.

In the next sentence (L10-11) the authors refer to proposal that TssA recruits the baseplate and primes sheath-tube extension. Which TssA are they referring to? It is very confusing. In my previous review of this manuscript, I recommended adopting the TsaB, TsaC, TsmA nomenclature to avoid such confusion. If the authors wish to continue to refer to the subunit as TssA, they should be clear on which type of TssA, as they make the distinction in the first place.

P4, L21: "First, we found that non-contractile sheath structures were assembled in the DtssM and other MC-BP deletion mutants, indicating that the initiation of sheath-tube assembly does not require a fully assembled MC-BP." OK, but where are these being primed. Are they initiating randomly in the cytosol, as there is no membrane anchored baseplate complex to assemble on? The formation of LPPs would suggest assembly is being primed from the membrane at one pole.

How does this occur?

P5, L4: use of the term 'repression' and at several places (particularly pages 7 and 9) rather than inhibition. Repression is usually used in the context of gene expression.

Results:

P5, L20 The authors must state that they are referring to the VipA-N3-sfGFP host here.

It is often not clear whether the authors are referring to the vipA+ or vipA-N3 host. The term 'wild type cells' is also used frequently, but it is not always clear whether the authors are referring to the vipA status, on the one hand, or the tssM, tssA or tagA status on the other.

P7, L11. TssA synthesis was induced....

P7, L14-15, is this the non-contractile tssM mutant? Please specify.

P8, L14: "Both wild-type-like and LPPS structures were formed, and competition assays show that DtagA.NTD mutant retained killing ability similar to DtagA cells"

This sentence is not clear. First, were these constructed in the non-contractile strain (I assume so)? Second in which mutants were the WT and LPPS-like structures formed, Δ tagA, Δ tagA.NTD or both?

I am surprised that cells containing LPPS secrete Hcp and kill prey? If they are non-contractile, how do they do this? Also, in mutants where there is no MC or BP, how does the tube-sheath fire and kill, as there is no membrane anchor point?

P8, L21: "Because of the inhibition of T6SS sheaths by TagA overexpression...."

Rephrase -inhibition of assembly of T6SS sheaths...

P9, L3-4: In what strain background are the authors deleting the TssA ImpA_N domain? Was the Δ tagA allele already present in the strain? if so, wouldn't it be easier to state at the outset that the Δ tagA allele was introduced into Δ tssA and Δ tssA.NTD mutants in vipA+ and vipA-N3 backgrounds?

P9, L9 (and elsewhere) Use of the term 'wild type' is confusing as I don't know if the authors are referring to the vipA status or to the tssM, tssA, tagA... status

P9, L8-11: "Since overexpression of TssM caused LPPS formation (in which strain background?), we tested its effect in the DtssA mutant. Unlike in wild type and DtssM cells, overexpression of TssM did not lead to LPPS structures in DtssA mutant but substantially stimulated formation of wild-type-like sheath-tube structures in both contractile and non-contractile cells."

P9, L13-14: "Cumulatively, these data suggest that TagA represses sheath assembly in the absence of TssA"

Avoid use of 'represses'.

Discussion:

The authors have not adequately explained how and where the sheath structures initiate assembly in the absence of the baseplate or a membrane complex. If there is no platform on which to polymerise the sheath, it could initiate at any point in the cytosol, yet LPPSs are observed, which implies an initiation site on a membrane. Please could the authors address this.

Reviewer #2 (Remarks to the Author):

The manuscript really improved after this round of revision. It is more detailed and with more

supporting data.

REVIEWER COMMENTS

Reviewer #1 (Remarks to the Author):

Stietz et al analyse the consequences of deleting T6SS subunit genes or overexpressing them in wild type cells and cells encoding an aberrant VipA (TssB) sheath subunit that assembles into sheaths that do not contract. The results are somewhat surprising given current models for T6SS function, and in some cases difficult to explain without a degree of speculation. Based on the interpretation of their observations, the authors propose a revised model to account for the roles of TssM, TssA and TagA in assembly of the tube-sheath complex.

This version of the manuscript is certainly easier to follow than the previous one, although there are still places where things are not crystal clear (which I indicate below).

Title:

What does 'termination' refer to in this context? Surely "...biogenesis and termination of T6SS sheath-tube assembly in *Vibrio cholerae*" would be more accurate?

Thank you for the suggestion and we have revised the title accordingly. New title reads:

"Assembling a needle without an anchor: functional interaction of TssA-TssM-TagA dictates biogenesis and termination of the type VI secretion system sheath-tube assembly in *Vibrio cholerae*" (added words underlined)

Abstract:

"Our results demonstrate that T6SS priming is not dependent on TssA, nor is its termination dependent on the distal end TssA-TagA interaction..."

Again, use of the term 'termination' needs to be more specific. The T6SS does not terminate. The authors mean termination of tube-sheath assembly.

We agree and have modified the statement as follows:

P2, L13-15 'Our results demonstrate that priming of the T6SS sheath-tube assembly is not dependent on TssA, nor is the assembly termination dependent on the distal end TssA-TagA interaction...'

Introduction:

P3, line 21 and P12, L6, the term 'sheath-needle' is not replaced by 'sheath-tube' as stated by the authors in their comment on P3, L11.

'Needle' is also used in isolation, i.e. Fig. 1 legend. This should be defined somewhere.

We thank the reviewer for pointing out those unintended omissions. We have now replaced 'sheath-needle' with 'sheath-tube' throughout the manuscript.

The word 'needle' was also replaced by 'tube' in Fig. 1a and by 'Tube-Spike' in Fig. 1b and Supplementary Fig. 1f (see below). The corresponding figure legends have been edited accordingly.

P3, L17-18, The authors state “The tip protein PAAR is critical in *Acinetobacter baylyi* ADP1 but dispensable in *Vibrio cholerae* for T6SS function” and yet it is clearly stated in the abstract to the reference cited by the authors (Shneider et al., 2013) that PAAR proteins are essential for the functioning of the *V. cholerae* T6SS.

Thank you for raising this point. We disagree with the published statement in the abstract of Shneider et al., 2013 that “PAAR proteins are essential for T6SS-mediated secretion and target cell killing by *Vibrio cholerae* and *Acinetobacter baylyi*”. Specifically, in Shneider et al., 2013 Fig 2 (shown below) and the corresponding text, the authors report that the double *paar* deletion mutant of *V. cholerae* showed reduced bacterial killing (at least ~100-fold decrease) (panel b) and 70% reduction in Hcp secretion (~3-fold decrease) (panel C), a hallmark for T6SS function. However, these results also demonstrate a ~1000-fold difference in *E. coli* survival between the double *paar* mutant and the T6S null mutant indicating T6SS of the double *paar* mutant is still active (panel b, red arrow highlighted). Hcp secretion of the double *paar* mutant was detectable by Coomassie blue staining, indicating plenty of Hcp proteins were secreted (panel C, red arrow highlighted). Notably, deletion of all three *paar* genes in ADP1 abolished Hcp secretion and T6SS killing resembling a T6SS null phenotype (panel a&c). Therefore, the statement in Shneider et al., 2013 that PAAR is essential for T6SS in both *V. cholerae* and ADP1 is inaccurate. Indeed, others have noted this issue. In a Trends in Microbiology review (Joshi et al., 2017), it reads “...Though not shown to be essential components of the baseplate, proteins containing repeating proline-alanine-alanine-arginine (PAAR) motifs cap the VgrG1-3 trimer and act to sharpen the T6SS spike complex [22]...”, citing the same (Shneider et al., 2013) article.

Figure 2 | PAAR proteins are required for full functionality of the T6SS in *Vibrio cholerae* and *Acinetobacter baylyi*. a, Recovery of viable *E. coli* MG1655 after co-incubation with *A. baylyi* ADP1 (wild type, WT) and its T6SS and PAAR genes knockout mutants. The following genes were inactivated in the mutants shown: T6S⁻, *aciad2688* to *aciad2694*; 2681⁻, *aciad2681*; 0051-52⁻, both *aciad0051* and *aciad0052*; 3 × P⁻, all three PAAR genes *aciad0051*,

forming units (c.f.u.) after co-incubation with *V. cholerae* 2740-80 and PAAR gene knockout mutants, which are labelled as follows: 105⁻, *vca0105*; 284⁻, *vca0284*; 2 × P⁻, *vca0105* and *vca0284*. Symb *** indicate deviations from the wild type with *P* values of 6×10^{-6} and 5×10^{-7} , respectively, for a sample size of 8. Error bars represent standard deviation. c, SDS-PAGE (sodium dodecyl sulphate-pol-

To clarify and avoid any misunderstanding, we have revised the statement to “The tip protein PAAR is essential for T6SS functions in *Acinetobacter baylyi* ADP1 but not strictly required in *Vibrio cholerae* ²¹.” (changed text underlined)

P4, L6-10, confusing. In the first sentence, the authors refer to TssA proteins that serve as chaperones as TssA or TsaC, whereas TssA proteins that remain baseplate associated are referred to only as TsaB. This would be OK if the authors stick to this definition. However, in the second sentence, where they are referring to TssA (and therefore, as far as the reader is concerned, they are referring only to TsaC), the authors provide references for the interactions of both types of TssA (the Planamente et al paper deals with TsaB). Moreover, they mention interactions of TssA with TssC, E, F, G, L, K, M, Hcp, VgrG and TagA. What TssA are they referring to, their definition (TsaC) or the broader definition (TsaB and TsaC)? TsaC does not interact with all of these proteins.

Thank you for pointing out this issue. We agree that because TssA in *E. coli* and *V. cholerae* and TssA1 in *P. aeruginosa* belong to difference classes, we should discuss them separately instead of mixing them together. To clarify, we have revised this section as follows (newly added information underlined).

“...The T6SS cluster of *V. cholerae* encodes two ImpA_N domain proteins, TssA and TagA, which belong to the TsaC and TsmA classes, respectively. It has been shown in *Escherichia coli* and *V. cholerae* that TssA (TsaC) could interact with the inner membrane proteins TssL and TssM, the baseplate TssEFGK-VgrG, TagA (TsmA) and subunits of sheath-tube VipB (TssC) and Hcp ²⁹⁻³¹ (Citation of Planamente et al., 2016 referring to TsaB was removed), playing a key role in priming and facilitating sheath-tube polymerization ²⁹⁻³¹. TagA (TsmA) is peripherally associated with the inner membrane and is believed to function as a stopper that terminates sheath extension upon interacting with TssA at the sheath distal end ^{31,33}. Deletion of *tagA* results in frequent formation of long and curved sheath-tube in both *E. coli* and *V. cholerae* ^{31,33,34}. The H1-T6SS cluster of *Pseudomonas aeruginosa* encodes a TsaB (TssA1) that is a baseplate component and interacts with TssK1, TssF1, ClpV1, and sheath-tube subunits VipA1(TssB1) and Hcp1 ³² (citation of Planamente et al., 2016 was added here). Despite of the structural and functional differences of these ImpA_N domain proteins, it seems to be common that they have multiple binding partners and their functions remain to be elucidated.”

In the next sentence (L10-11) the authors refer to proposal that TssA recruits the baseplate and primes sheath-tube extension. Which TssA are they referring to? It is very confusing. In my previous review of this manuscript, I recommended adopting the TsaB, TsaC, TsmA nomenclature to avoid such confusion. If the authors wish to continue to refer to the subunit as TssA, they should be clear on which type of TssA, as they make the distinction in the first place.

Please see the response above. In addition, we also clarified the use of TssA (TsaC) vs TsaB in other sections of the manuscript (mentioned in the previous comment) and in Fig. 6a and figure legend (please see below).

Description added (underlined) to Figure Legend: Fig 6a. Note that TssA in VC and EC belong to the TsaC functional class, while TssA1 in PA belongs to the TsaB functional class, thus denoted with asterisk. TagA in VC and EC belong to the TsmA class.

P4, L21: “First, we found that non-contractile sheath structures were assembled in the *DtssM* and other MC-BP deletion mutants, indicating that the initiation of sheath-tube assembly does not require a fully assembled MC-BP.” OK, but where are these being primed. Are they initiating randomly in the cytosol, as there is no membrane anchored baseplate complex to assemble on? The formation of LPPs would suggest assembly is being primed from the membrane at one pole. How does this occur?

Thank you for the comments. These sheath-tube structures, while ultimately becoming LPPs, appear to prime and originate randomly throughout the cell, rather than being limited to the pole area. We have added this statement below and included new data as Supplementary Fig. 1g, h and Supplementary Movie1 (see below).

P6, L12-14 “Notably, initiation of these LPPs structures is not limited to the pole area but appears to occur randomly throughout the cell (Supplementary Fig. 1g, h and Supplementary Movie 1).”

New data include the analysis of 30 non-contractile VipA-N3-sfGFP polymerizing sheaths assembled in the *ΔtssM* deletion mutant. Without an intact membrane complex-baseplate anchor, at least some non-pole initiating sheath-tube structures seem to be subject to relocation to allow continual polymerization of sheath-tube structures, leading to LPP formation (see Supplementary Fig. 3d).

Figure Legend. **g**. Schematic indicating the initiation point of 30 non-contractile VipA-N3-sfGFP sheaths (orange dots) formed in the $\Delta tssM$ deletion strain. Dashed lines demarcate cell pole areas. The percentage of sheath initiations observed in each cell area is indicated at the top. **h**. Priming region of non-contractile sheaths polymerizing in the $\Delta tssM$ deletion strain. Cropped cells correspond to VipA-N3-sfGFP signal in grayscale at time 0 of sheath polymerization. Yellow dashed lines show cell outlines, orange arrows indicate sheath initiation points. Examples of sheath polymerization in non-contractile $\Delta tssM$ are included as Movie 1.

Movie Legend: **Movie 1. Initiation and polymerization of non-contractile sheaths in the $\Delta tssM$ strain.** Video plays VipA-N3-sfGFP non-contractile sheaths polymerizing in the $tssM$ deletion strain, 4 examples are shown. Each example corresponds to a 5 min time-lapse acquisition and plays at a frame rate of 10 sec per frame. Scale bar 1 μ m. A merge of GFP and brightfield channels is shown.

P5, L4: use of the term ‘repression’ and at several places (particularly pages 7 and 9) rather than inhibition. Repression is usually used in the context of gene expression.

The term ‘repression’ has been replaced by ‘inhibition’ throughout the manuscript.

Results:

P5, L20 The authors must state that they are referring to the VipA-N3-sfGFP host here.

Thank you. We have now added “non-contractile” to the statement below

“Surprisingly, long and often pole-to-pole sheath structures were found in all non-contractile MC-BP mutants...” (added text underlined)

It is often not clear whether the authors are referring to the *vipA+* or *vipA-N3* host. The term ‘wild type cells’ is also used frequently, but it is not always clear whether the authors are referring to the *vipA* status, on the one hand, or the *tssM*, *tssA* or *tagA* status on the other.

Thank you for this point. To improve clarity, we have done a systematic review on all “wild type” used in this manuscript and replaced with more specific description wherever needed.

The term wild type was replaced in the following sentences:

P5, L7 TssA+

P6, L9 TssM+

P7, L16 VipA-N3-sfGFP

P8, L10 and L17 TssA+

P8, L23 TagA+

P9, L7-9 “T6SS sheath formation was significantly increased (Fig. 5a, d-g, Supplementary Fig. 4a, Supplementary Movie 5). However, there were still 20-fold (contractile) and 6-fold (non-contractile) less than the corresponding parental strains and killing ability was not restored in the T6SS contractile cells (Fig. 5a, d-g, Supplementary Fig. 4a, Supplementary Movie 5).” (added text underlined)

P9, L16 TssM+

P7, L11. TssA synthesis was induced....

Thank you. We have revised accordingly.

“Interestingly, induction of TssA synthesis in the non-contractile wild type cells doubled the sheath length and stimulated the formation of LPPS...” (revised text underlined).

P7, L14-15, is this the non-contractile tssM mutant? Please specify.

Yes, it is the non-contractile *tssM* mutant. We have added “non-contractile” to the statement below accordingly:

“Surprisingly, when induced in the non-contractile $\Delta tssM$ mutant, expression of TssA abolished sheath-tube formation”

P8, L14: “Both wild-type-like and LPPS structures were formed, and competition assays show that DtagA.NTD mutant retained killing ability similar to DtagA cells”

This sentence is not clear. First, were these constructed in the non-contractile strain (I assume so)?

Second in which mutants were the WT and LPPS-like structures formed, $\Delta tagA$, $\Delta tagA$.NTD or both?

I am surprised that cells containing LPPS secrete Hcp and kill prey? If they are non-contractile, how do they do this? Also, in mutants where there is no MC or BP, how does the tube-sheath fire and kill, as there is no membrane anchor point?

This is likely a misunderstanding. They are contractile cells. To clarify, we have revised the following paragraph (newly added text underlined).

“We next imaged contractile sheath formation in the $\Delta tagA$ and the $\Delta tagA^N$ mutants, the latter lacking the ImpA_N domain. In these VipA-sfGFP contractile cells, both wild-type-like and LPPS structures were formed, and competition assays show that $\Delta tagA^N$ mutant retained killing ability similar to $\Delta tagA$ cells (Fig. 4f, g, Supplementary Movie 4).”

P8, L21: "Because of the inhibition of T6SS sheaths by TagA overexpression...."

Rephrase -inhibition of assembly of T6SS sheaths...

Done. The statement now reads: "Because of the inhibition of assembly of T6SS sheaths by TagA overexpression"

P9, L3-4: In what strain background are the authors deleting the TssA ImpA_N domain? Was the $\Delta tagA$ allele already present in the strain? if so, wouldn't it be easier to state at the outset that the $\Delta tagA$ allele was introduced into $\Delta tssA$ and $\Delta tssA.NTD$ mutants in $vipA^+$ and $vipA-N3$ backgrounds?

The TssA ImpA_N domain deletion was constructed in wild type contractile background. To clarify, we have revised this sentence:

P9, L10-13: "In addition, deleting the TssA ImpA_N domain alone reduced sheath-tube levels similar to that of the $\Delta tssA \Delta tagA$ mutant in contractile cells, and introducing the TagA ImpA_N deletion to the TssA ImpA_N deletion mutant did not lead to any further increase (Fig. 5b, e, Supplementary Fig. 4a, Supplementary Movie 6)..." (added text underlined)

P9, L9 (and elsewhere) Use of the term 'wild type' is confusing as I don't know if the authors are referring to the $vipA$ status or to the $tssM$, $tssA$, $tagA$... status

Wild type was replaced by $TssM^+$ in "Unlike in $TssM^+$ and $\Delta tssM$ cells..." here (P9, L16) and in P6, L9. As we have stated earlier, we have also changed wild type to more specific description wherever appropriate throughout the text.

P9, L8-11: "Since overexpression of TssM caused LPPS formation (in which strain background?), we tested its effect in the $\Delta tssA$ mutant. Unlike in wild type and $\Delta tssM$ cells, overexpression of TssM did not lead to LPPS structures in $\Delta tssA$ mutant but substantially stimulated formation of wild-type-like sheath-tube structures in both contractile and non-contractile cells."

As requested, we have added strain information in the statement below.

"Since overexpression of TssM caused LPPS formation in both contractile and non-contractile cells (Fig. 2a, d), we tested its effect in the $\Delta tssA$ mutant. Unlike in $TssM^+$ and $\Delta tssM$ cells (Fig. 2d), overexpression of TssM did not lead to LPPS structures in $\Delta tssA$ mutant but substantially stimulated wild-type-like sheath-tube structures in both contractile and non-contractile cells"(newly added text underlined)

P9, L13-14: "Cumulatively, these data suggest that TagA represses sheath assembly in the absence of TssA"

Avoid use of 'represses'.

The term 'represses' was substituted by 'inhibits'.

Discussion:

The authors have not adequately explained how and where the sheath structures initiate assembly in the absence of the baseplate or a membrane complex. If there is no platform on which to polymerise the sheath, it could initiate at any point in the cytosol, yet LPPSs are observed, which implies an initiation site on a membrane. Please could the authors address this.

Thank you for the comments. We believe we have touched upon some points on LPPS formation in the response above. We have also revised the text in the results and discussion below:

Results:

P6, L12-14 “Notably, initiation of these LPPS structures is not limited to the pole area but appears to occur randomly throughout the cell (Supplementary Fig. 1g and Supplementary Movie 1).”

Discussion:

P11, L3-5 “How is the LPPS generated? We noticed that sheath-tube structures, while ultimately becoming LPPSs, appear to originate randomly throughout the cell, rather than being limited to the pole area (Supplementary Fig. 1g and Supplementary Movie 1).”

P11, L14-19 “However, it remains elusive how LPPS structures could initiate in defective MC-BP mutants. Because LPPS is most abundant in the $\Delta tssM$ mutant while mutants lacking individual BP components exhibit reduced formation of LPPS (Fig 1B), the theoretically intact BP in the $\Delta tssM$ may initiate sheath-tube polymerization independent of the MC at higher efficiency than in the individual BP mutants. Future work on visualization of LPPS using Cryo-Electron Tomography would provide valuable insights on assembly in those MC-BP mutants.”

We share the reviewer’s curiosity that it remains intriguing regarding how exactly the sheath structures initiate assembly in those mutants. We believe direct visualization using Cryo-ET is perhaps the best way to address this question and have initiated collaborative work on this. Nonetheless, this study not only highlights the important roles of TssA-TssM-TagA in biogenesis of T6SS sheath-tube assembly but also enable us to use a panel of MC-BP mutants to dissect the contribution of each component to T6SS assembly, which likely leads to more in-depth understanding how each component works using the conditions we established in future studies.

Reviewer #2 (Remarks to the Author):

The manuscript really improved after this round of revision. It is more detailed and with more supporting data.

Thank you very much.

REVIEWERS' COMMENTS:

Reviewer #1 (Remarks to the Author):

The authors have satisfactorily addressed my comments and queries.

REVIEWERS' COMMENTS:

Reviewer #1 (Remarks to the Author):

The authors have satisfactorily addressed my comments and queries.

We thank the reviewer for all the feedback.